# Spatial localisation meets biomolecular networks

Govind Menon [1] & J. Krishnan [1,2✉]

Spatial organisation through localisation/compartmentalisation of species is a ubiquitous but poorly understood feature of cellular biomolecular networks. Current technologies in systems and synthetic biology (spatial proteomics, imaging, synthetic compartmentalisation) necessitate a systematic approach to elucidating the interplay of networks and spatial organisation. We develop a systems framework towards this end and focus on the effect of spatial localisation of network components revealing its multiple facets: (i) As a key distinct regulator of network behaviour, and an enabler of new network capabilities (ii) As a potent new regulator of pattern formation and self-organisation (iii) As an often hidden factor impacting inference of temporal networks from data (iv) As an engineering tool for rewiring networks and network/circuit design. These insights, transparently arising from the most basic considerations of networks and spatial organisation, have broad relevance in natural and engineered biology and in related areas such as cell-free systems, systems chemistry and bionanotechnology.

[1] Chemical Engineering, Centre for Process Systems Engineering, London, UK. [2] Institute for Systems and Synthetic Biology, Imperial College London, London, UK. ✉email: j.krishnan@imperial.ac.uk

Networks are pervasive across a range of natural and engineered systems, and the analysis of networks is a basic way of dissecting, elucidating and engineering these systems. Indeed, network analysis represents a basic way of thinking about a broad variety of complex systems, which might, on the face of it, have little in common. Biomolecular networks are a central ingredient of systems biology, synthetic biology, biological engineering and systems chemistry. Complex biomolecular networks are central to how information is processed in these systems, and how cellular life is organised and maintained. Despite the wide diversity of networks, they are ultimately comprised of key motifs and modules, which are widely observed, and studying these recurrent building blocks has been a fruitful way to study basic aspects of network function.

Spatial organisation is another fundamental aspect of living systems, seen at multiple levels spanning cellular, tissue, organ and organism levels. At the intracellular level, the ubiquitous presence of localisation/ compartmentalisation is a basic feature of bacteria and eukaryotes, with spatial organisation being directly exploited in evolution. There are pervasive ingredients encountered at multiple levels, notably the localisation of components and the disparity in transport rates between components (some components being highly diffusible and others weakly diffusible). This is especially evident at the cellular and tissue level.

Understanding the interplay of spatial organisation and biochemical networks is therefore fundamental to the understanding and engineering of biological systems, with further relevance to cell-free systems. There are a variety of examples testifying to the broad relevance of this theme (Fig. 1a).

Systems/synthetic biology presents numerous examples of the confluence of weakly/non-diffusible, highly diffusible and localised entities (discussed in detail in Supplementary Information). Cellular examples span bacteria and eukaryotes (C. Elegans, fungi, mammalian cells), different pathways (e.g. MAPK, Calcium signalling) with different roles/outcomes: gradient generation, polarity generation, spatial positive feedback and switching, membraneless compartments[1–6]. Population-level examples include multifunctional circuits in development, localisation of metabolic pathways in different cells/spatial zones (for instance, determining tissue-level insulin response)[7,8]. Synthetic biology examples include compartmentalisation of circuits in cell-free systems, artificial and natural cells, reconstituting circuits in cells, engineering cellular decision-making for tissue engineering and creation of synthetic population mimics for programming multicellular structures[9–14].

The above examples clearly indicate the need for an in-depth understanding of the interplay between space and biochemical/ biomolecular networks, which is broad-based and transcends the individual context. In this paper we develop a systems approach, systematically accounting for network features and spatial aspects (in particular, localisation) to achieve this. In so doing, we reveal multiple facets of this interplay, and address both network-centric and space-centric issues: (i) What does spatial organisation contribute to the behaviour of networks, and our understanding of processes in terms of networks? (ii) How does the presence of spatial organisation affect data-driven inference and modelling of processes through networks? (iii) What does the consideration of network features contribute to our understanding of basic spatial aspects of interest such as gradient-sensing and pattern formation (and spatial information processing)? (iv) How can spatial organisation be employed in the engineering of biomolecular networks?

## Results

Some basic aspects of spatial organisation in networks, arising from the interplay of local and global entities, have been studied in the literature, from various perspectives, including the capacity to generate patterns, building from Turing's classic work[15], for e.g. see ref. [16–18] and the behaviour of network modules with elements of contrasting diffusivities[19]. In the present study, a crucial new ingredient is the presence of spatial localisation, a widespread ingredient in biological systems. We focus on this, by examining the interplay of localised with local and global entities in networks.

We examine multiple facets of the interplay of biochemical/ biomolecular networks and space/localisation (see schematic in Fig. 1b, c) from two complementary perspectives (see Methods for details). We consider three types of spatial characteristics of species: local (weakly/non-diffusible), global (highly diffusible) and localised (species localised at designated spatial regions). A spatial network description involves the specification of the spatial domain (and boundaries) and spatial characteristics of nodes, in addition to the network interactions. We employ partial differential equation (PDE) models in a 1-D spatial domain (sufficient for this study) with appropriate boundary conditions (no-flux, periodic). A network is comprised of nodes (of interconverting species) and interactions between nodes. Unless otherwise mentioned, the species (active/inactive forms) constituting a node are assumed to have the same spatial characteristic (either localised, highly diffusible (global), or weakly/ non-diffusible (local)). Our study focusses on a broad class of candidate spatial networks, obtained as follows. One exploration starts by examining networks, with different spatial characteristics of nodes. To do this we examine a variety of three-node motifs which encompass a wide range of relevant behaviour[20], imposing different spatial characteristics of nodes systematically. We build on this to also examine the spatial charactersitics of the interaction (which may involve other species implicitly) by again associating them with the aforementioned spatial characteristics. In this manner, the use of realistic and representative spatial organisation in network motifs allows us to access and analyse networks which are not limited in their spatial complexity. This approach is more focussed and fruitful than scanning motifs of larger sizes and focussing exclusively on the spatial characteristics of nodes. A complementary perspective starts from the spatial classes and involves examining how a given network is distributed between the three spatial classes. This involves building up spatial networks with the different classes as a basic vantage point. While these perspectives essentially overlap for simple networks and simple localisation patterns, they provide distinct vantage points from which to examine and construct more complex spatially organised networks.

Our results below are underpinned by a broad exploration of the impact of localisation in networks (as outlined above). We present representative and striking features of the effect of localisation as follows: (i) basic effects of localisation, which are observed even in very simple networks (ii) effects of localisation on networks exhibiting specific characteristics (e.g. bistability, oscillations). We subsequently present results on (iii) pattern-forming capabilities of such networks (iv) the impact of localisation on network inference and (v) a selection of cases from an engineering perspective. We employ a combination of basic networks (illustrating general points, broadly relevant), specific networks (exhibiting noteworthy behaviour), and additional exemplar cases (demonstrating specific applications).

**The effect of spatial localisation on networks.** We first address the question: how and in what types of ways can spatial localisation affect network behaviour and function?

Localisation, by itself, introduces multiple new features: gradient creation, dependence on diffusivity and location relative to boundaries. The most basic effect of having localised entities in

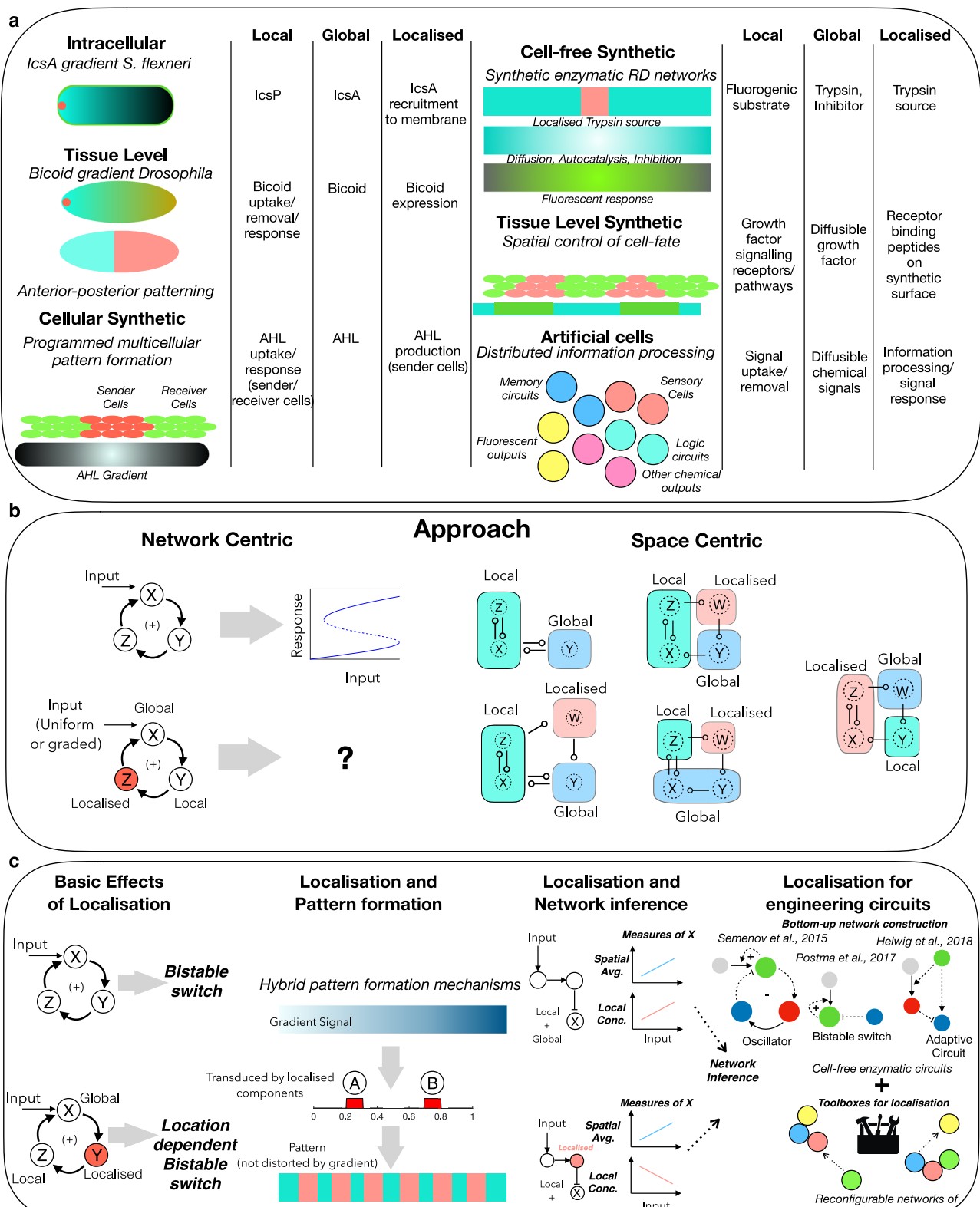

**Fig. 1 Schematic summary of the focus of the paper, the ingredients, approaches and types of insights. a** Schematic summarising the central role of the interplay between local (non-diffusible/slow diffusing), global (diffusible/fast diffusing), and localised components, in multiple contexts of biochemical information processing in natural and engineered biology and at their junction (related references: Robbins et al.[1], Durrieu et al.[49], Basu et al.[50], Semenov et al.[9], Li et al.[14], Joesaar et al.[13]). **b** Schematic illustrating the two complementary approaches used in our study, bringing together networks and spatial classes. See 'Methods' for details. **c** Schematic summarising the types of insights emerging from the study, elucidating the role of localised components in spatial and temporal information processing by biochemical networks. combining basic effects, the impact on pattern formation, the hidden role in network inference and the potential for engineering synthetic circuits (the results arising from the study are summarised in Fig. 10).

a network is that it introduces steady state gradients to other species in the network (whose diffusivity is at neither extreme). We examine this in a two-node feedback motif by localising one of the interactions. This results in gradients for both nodes, which may be commensurate or opposing, which depends on whether the non-localised interaction is activating or inhibiting (Fig. 2b). In general, the relative nature of gradients depends on the nature of the non-localised interaction, rather than the overall behaviour of the circuit: this is clearly exemplified by the case of feedback circuits.

Another basic feature that emerges from localisation is the effect of boundaries (described by no-flux boundary conditions in this instance) of the spatial domain. Localised regulation (activation/inhibition) causes characteristics of the steady state response to depend on position of localisation relative to boundaries. This is exemplified by the case of a single node (moderate diffusivity) with a localised activating signal (Fig. 2c, d). Furthermore, two different measures of the output (local concentration at input location and spatial average concentration) can exhibit opposite trends in response to change of input location relative to boundaries. The former is amplified as the input location moves closer to a boundary, while the latter is

attenuated. The effect of a change of localisation position can be accentuated for moderate diffusivities, while it is minimal for low and high diffusivities. An associated insight is that changing species diffusivity for fixed localisation, can result in a non-monotonic response.

What are the consequences for the qualitative behaviour of basic information-processing motifs? We build on the above observations to examine two basic information processing motifs exhibiting distinct characteristic behaviour—a two-node positive feedback motif exhibiting bistability, and a three-node negative feedback motif exhibiting oscillations. We see that the above effects can produce qualitative changes in the dynamical characteristics of such motifs. For instance, in the bistable motif with one of the nodes localised and the other diffusible, the location of the localised element can affect bistable characteristics—with locations closer to a boundary favouring bistability (Fig. 2f). Similarly, the location of localised nodes in an oscillator motif with one of the nodes localised and the others diffusible can impact oscillatory behaviour—with locations closer to a boundary favouring oscillations (Supplementary Fig. 2). Some of these aspects can be understood analytically (see Supplementary Notes 1.2, 1.3). Non-trivial diffusivity-dependent behaviour (a consequence of localizaion) manifests itself

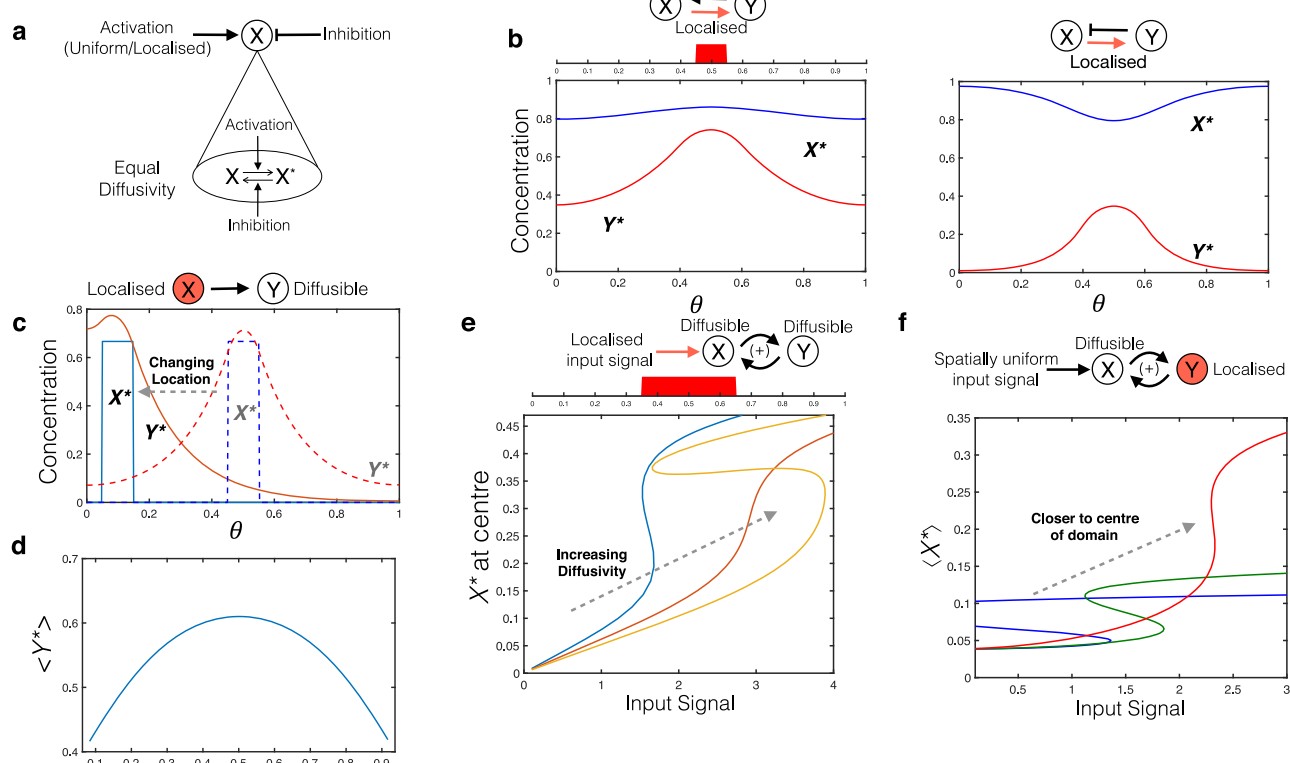

**Fig. 2 Basic effects of localisation: location and diffusivity-dependent behaviour.** In all figures that follow, $\theta$ denotes the spatial co-ordinate, and the concentration of the species plotted is the active form of the node in the associated motif. A shaded node in a motif denotes a localised node and a coloured arrow denotes a localised interaction. **a** Schematic of a single node highlighting how activation or inhibition is implemented. **b** When an interaction is localised in a feedback motif, the resulting gradients of species can be commensurate or opposing depending on the nature of the non-localised interaction (blue curves denote $X^*$, red curves denote $Y^*$). **c** Altering the location of localisation of input node X introduces location dependence of the steady state of a moderately diffusible output node Y, without affecting X itself. A higher level of output is attainable locally (for a fixed input level),when the localisation is closer to a closed boundary (solid curves denote the case where localisation is close to the boundary, dashed curves, the case where localisation occurs in the middle of the domain). **d** This location dependence is also reflected in other measures such as the spatial average of Y (hereafter spatial average is denoted by < >). In contrast to (**c**), the maximum value of the average occurs when X is localised in the centre of the domain. **e** Diffusivity-dependent behaviour: The bistable behaviour of a two-node positive feedback motif with localised input signal is affected by the (equal) diffusivity of the nodes. The system exhibits bistability for low and higher diffusivities, and loses it at intermediate diffusivity. **f** The same motif, with one of the nodes localised, can exhibit location dependent bistable behaviour, with locations closer to a boundary favouring bistability. In (**e**) and (**f**) curves with different colours denote different bifurcation curves as the diffusivity is varied.

clearly in these motifs. With a single localised element, we see that the bistable/oscillatory behaviour may be seen for both high and low diffusivity, but may be absent for intermediate diffusivity (Fig. 2e, Supplementary Fig. 5c).

The previous results involve a single localised element (input/node/interaction). We now discuss a basic aspect associated with localisation at a multiplicity of locations, introducing qualitative distortions. Localising two interactions in a positive feedback bistable circuit can destroy bistability simply by weakening the interactions and consequently the feedback. Another example of a basic distortion of qualitative behaviour is seen when the two nodes of an incoherent feedforward adaptive (homoeostatic) circuit are localised at different locations: such a scenario by itself destroys the adaptation/homoeostatic response of these circuits (even with respect to uniform stimuli: see Supplementary Note 1.4).

Localisation enables new types of information processing characteristics which may be otherwise inaccessible. The new qualitative characteristics introduced can be seen by contrasting the behaviour of the actual spatially distributed system, with the ODE model of the interactions (which assumes all entities in the same location). This is illustrated by the following two cases. A negative feedback circuit with interactions localised at two different locations is capable of exhibiting oscillations (Supplementary Fig. 5a)—something impossible in the ODE model (see Supplementary Note 1.5). Interestingly, in this case, oscillatory behaviour is seen only for intermediate diffusivity, in contrast to the above. Similarly for the three-node motif in Fig. 3a, the ODE network description admits bistability (but no higher order multistability) whereas the distributed system with two of the interactions localised is capable of exhibiting tristability (Fig. 3b–d). The tristability is associated with two steady states where one of the species is essentially uniform (and high) and the other localised (and low) (analogous to the ODE steady states), while the other exhibits graded profiles for both species, representative of a polarised state. Both these examples point to the removal of basic constraints by distributing the circuit.

Experimental studies in the *C. elegans* zygote point to a mechanism of polarisation which echoes the basic motif studied above[21]. Here the role of localised activation considered above is played by the localised recruitment of ParA and ParB (the inhibitory nodes of the motif) to the membrane.

Localisation can be used as a control mechanism for translating network behaviour into spatial outcomes. We highlight a non-trivial instance of such a possibility, focussing on bistable circuits. We show how localisation at multiple locations in the context of bistable switch circuits can result in the maintenance of different steady states in the regions between the localised sub-domains by the trapping of fronts at these locations (Fig. 3e, f). We demonstrate that a localised node inhibiting the two nodes of a mutual inhibition bistable circuit at two different locations facilitates the maintenance of two distinct steady states in the intermediate regions between these locations (the two locations are placed symmetrically in a periodic domain for specificity). This also demonstrates how switching between the steady states can be established independently in each region, by a simple graded input signal. The effect of localisation is both to segregate the domain, and 'control' the bistable circuit to enable distinct steady states in different regions.

**Localisation and spatial patterns**. What is the effect of co-localising both forward and reverse interactions of a node? In contrast to localisation of only a single reaction in a node of a network (e.g. localising a species upstream regulating it) if the opposing reaction is also localised at the same location, the gradient-inducing effect of localisation is neutralised, allowing

the node to exhibit a homogeneous steady state profile. If localisation is present only for this node (as described above), the network exhibits a homogeneous steady state, identical to the ODE. Note that the output species at this node is present everywhere (non-localised): see Supplementary Note 2.1. This observation applies to all the motifs under consideration. Strikingly, we observe that for certain motifs, it is possible that, depending on the choice of localised elements, this homogeneous steady state may lose stability.

Localisation can actually be a trigger for pattern formation in such cases. We demonstrate this localisation-induced instability in the case of an activator-inhibitor motif with moderately diffusible nodes (Fig. 4). With the reactions of the inhibitor node (Z) localised in the same location, we see that the homogeneous steady state (corresponding to the ODE steady state) is unstable, and the system evolves to a highly non-homogeneous steady state (see Fig. 4d where the uniform steady state can be unstable in contrast to 4c). Simulations indicate that there are multiple inhomogeneous asymptotic steady states. Furthermore, we see that this destabilisation does not require any difference in diffusivity of the nodes. We emphasise that, in the absence of localisation, the system has a unique uniform steady state which is locally stable, as can be seen directly from computational results and analysis. Furthermore, the observed instability arises only when the region of localisation is below a certain size. Taken together, this shows how localising basic interactions by themselves can generate patterns which can be built upon by the other network interactions. Even in parameter regimes where the homogeneous steady state is stable, the system can have co-existing stable inhomogeneous steady states (and even stable oscillatory states).

Simulations of the PDE indicate that the number and nature of inhomogeneous steady states can depend on the position of the localisation relative to boundaries. For instance, localisation in the centre of the domain (dividing the domain into identical halves: see Fig. 4e can give rise to a pair of symmetry-broken, inhomogeneous steady states (arising from a subcritical pitchfork bifurcation in a compartmental model analogue), with output level high in one half and low in the other. In addition, symmetric inhomogeneous stable steady states can also co-exist. Other types of destabilization (arising from a transcritical bifurcation) are seen when the localisation is not symmetrically placed (e.g. it is adjacent to a boundary: Fig. 4d. In the symmetric case, combinations of both transitions may be observed. The Supplementary Notes 2.2, 2.3 discuss these aspects further.

We now discuss how the confluence of localised and global elements can generate hybrid pattern-forming systems. Localisation combined with a global element allows the system to pick up information from different parts of the domain and feed it into a pattern-forming motif, without directly introducing distortions to the pattern from the localised element. We demonstrate an instance of this by considering two localised elements transducing information from the environment and feeding into a global element and subsequently a pattern-forming motif (Fig. 5a–d). This allows for the confluence of two appealing features with regard to the external signal, (1) by suitable construction, it is possible that homogeneous signals do not trigger patterns but gradient signals can and (2) the nature of the pattern is not spatially distorted by the localisation.

Positional information (with an externally imposed gradient) and Turing type pattern formation are two central mechanisms in developmental biology. The combination of these mechanisms is a current theme of interest, and the different configurations in which the two can combine and where these are encountered have been studied, with positional information upstream or downstream or in parallel with a Turing mechanism ([22],[23]). The

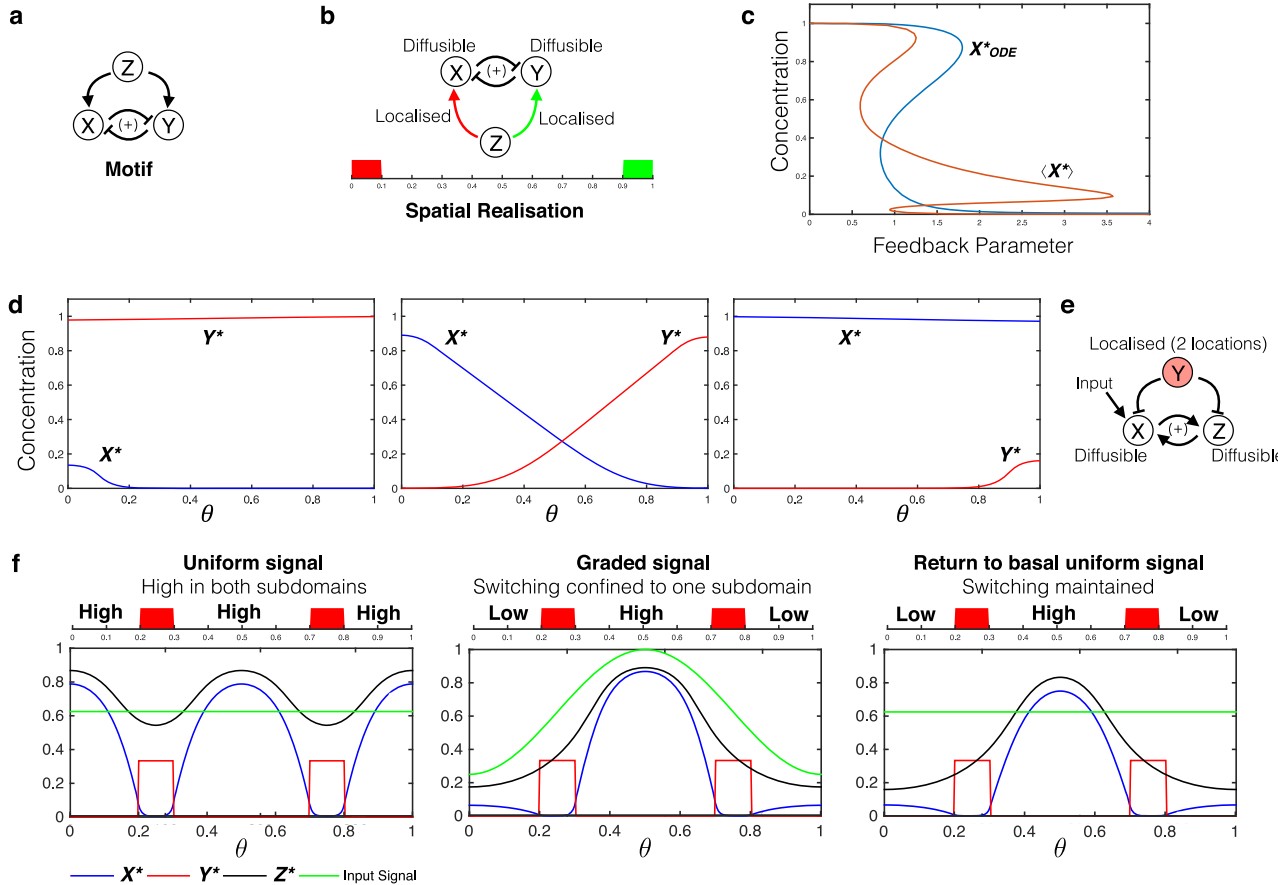

**Fig. 3 Localisation can introduce new qualitative behaviour. a** Bistable to tristable transition: motif under consideration, mutual inhibition between X and Y. **b** The spatially distributed realisation considered here involves the regulation of diffusible nodes X and Y by Z with the regulation localised at opposite ends of the domain. **c** Bifurcation diagram, with feedback strength of Y on X as bifurcation parameter. The corresponding ODE model exhibits bistability (blue curve) while tristability is seen in the distributed system (red curve depicting the spatial average). **d** The three stable, inhomogeneous steady state profiles are shown: in addition to two steady states where X or Y dominate, the system has a third steady state where they co-exist, each dominating in a particular region. Here blue and red curves denote $X^*$ and $Y^*$ respectively. **e** Spatial switching: motif under consideration, mutual activation (giving bistability) between X and Z (moderately diffusible); both inhibited by node Y, which is localised at two symmetric locations in a periodic domain. **f** Localised inhibition by Y divides the domain into two parts, and by trapping of fronts, and allows essentially different steady states of the X–Z bistable motif to be maintained in the two parts of the periodic domain (see left and right panels: both regions have high levels of X and Z (ON) in the left panel, while in the right panel, one region has low levels of X and Z (OFF)). Furthermore, the switching of steady states confined to specific sub-domains can be achieved by a graded input signal, and this switching remains when the gradation of signal is removed.

model just described is an example of a hybrid model which combines both these aspects in an essentially different way: the gradient is essential for pattern formation, while the pattern itself results from an instability (and depends on initial conditions: see Supplementary Note 3). This approach also allows for the construction of more complex pattern-forming elements which transduce different types of information from the environment, without distorting the resulting pattern.

Localisation can also be a basis for pattern selection. We demonstrate another application of combining localised and global elements, in the context of feedback regulation of pattern formation (see Fig. 5f, g). We consider an activator-inhibitor motif where an added negative feedback regulation is essential to facilitate patterns—stronger homogeneous feedback enables pattern formation (which is triggered when a perturbation above a certain threshold is introduced). Introducing localisation by incorporating an additional pathway with combination of localised and global elements in this feedback can help bias the choice of patterns to certain phases, without any distortions introduced to the pattern. This results in patterns which have a maximum, coincident with the location of the localised element,

while preventing patterns with a minimum here. The essential insight follows directly from the understanding of the homogeneous feedback (see Supplementary Note 3.1), illustrating an appealing feature of combining localisation and feedback to both facilitate and specify patterns.

**Spatial organisation and network inference**. A basic ingredient in building network models in natural contexts is that of measurement and making inferences about networks from data. The combination of networks, spatial organisation, and various types of measurements introduces certain complexities with regard to measurement and inference, even in cases involving ODE models where spatial organisation is not the focal point. We now examine multiple aspects of this. We first note that typical measurements (concentrations associated with a particular node) used to build ODE models are broadly of two different kinds—a local measurement (at a particular location), or an average measurement. Note that in many experimental contexts, these measurements of a node may be indirect, relying on downstream readouts. This dichotomy in measurement (localised vs. average) is also encountered in cell population-level studies.

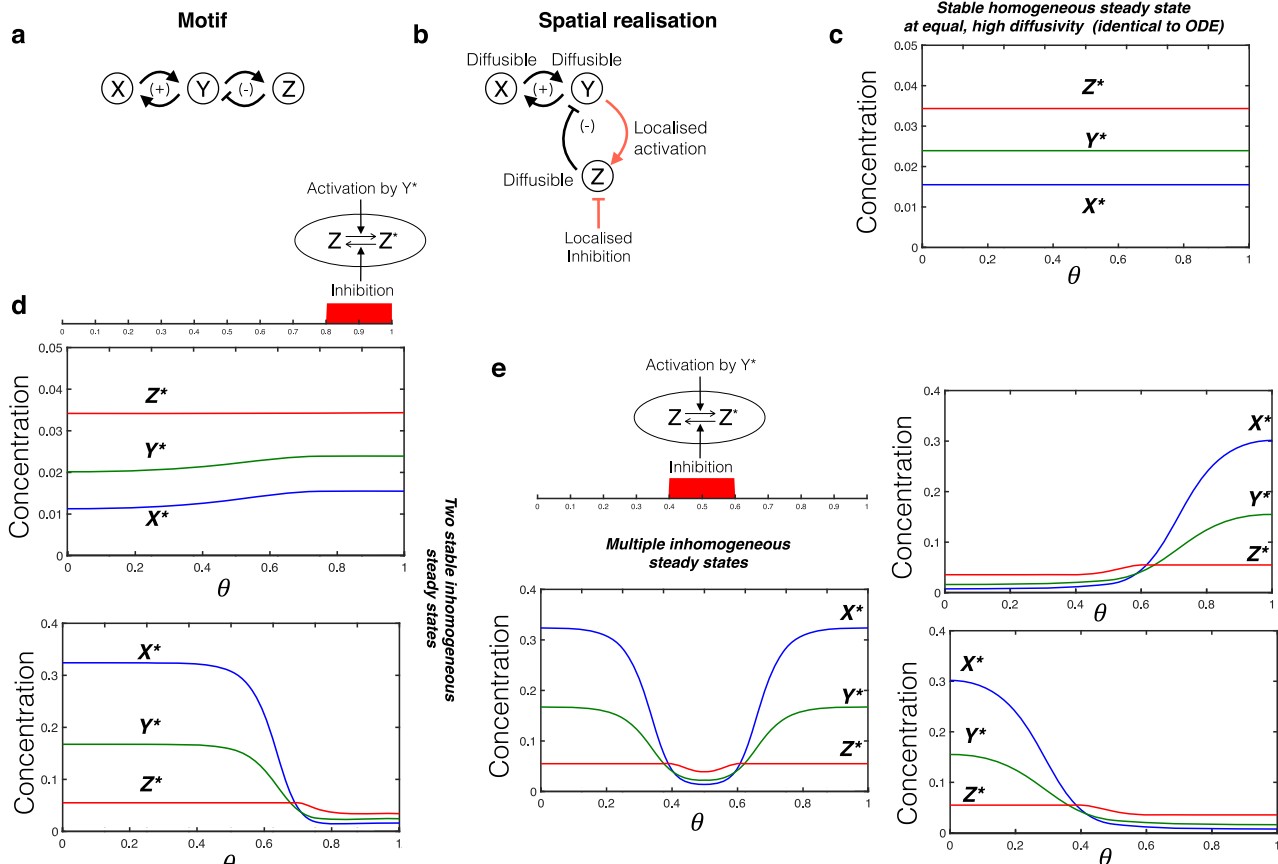

**Fig. 4 Localisation-induced instability. a** Motif under consideration: activator-inhibitor type motif where X is the activator and Z the inhibitor. **b** Schematic of spatial realisation: both forward and backward reactions at node Z are co-localised (the spatial domain has closed boundaries). **c** This localisation allows the system to have a homogeneous steady state, exactly equivalent to the ODE steady state. However, the homogeneous steady state may not be stable, and the system can exhibit multiple stable inhomogeneous steady states. Blue, green and red curves denote $X^*$, $Y^*$ and $Z^*$ respectively in (**c–f**). **d** When the localised sub-domain is adjacent to a boundary, the system can have two co-existing inhomogeneous steady states, one where X and Y output are high outside the localised sub-domain, and another where they are low. **e** With the localised sub-domain located symmetrically with respect to boundaries, the system can exhibit multiple inhomogeneous steady states, including two symmetry-broken steady states where the X and Y output are high in one part of the domain and low in the other (right column), as well as symmetric steady states (left column).

Different measures of the same node (local and spatial average) can be quantitatively different when the network contains localised elements as seen in the case of a simple signal transduction cascade with a single localised element (see Fig. 6a–c). Such quantitative differences may be accentuated and reflect as qualitative differences, in the presence of other non-linearities in the network such as thresholds (Fig. 6c).

An additional aspect of understanding spatially distributed networks is the relationship between measurements, behaviour of nodes and the network picture as captured by an ODE description. We make two points in relation to this: (1) it is possible that one type of measurement may give an input–output dose response curve close to the ODE description while the other type of measurement may not (see Fig. 6b in comparison to Fig. 6a), (2) in the same network, the behaviour of some nodes (in terms of a particular type of measurement) may conform to the ODE while others do not (see Fig. 6b, c showing this dichotomy for nodes Y and Z, with spatial averaged measurements).

These dichotomies have important consequences for the use of such network ODE models, noting that depending on the application context, particular nodes, and/or output measures may be especially important. It can result in either significant deviations in the behaviour of some (unmonitored) nodes, or in incorrect inferences drawn as to the source of observed behaviour.

In many networks, only some nodes are measurable. The presence of localised elements can introduce new pitfalls in making inferences about the network using such partial measurements. For instance, we have already seen how the location of a localised element relative to boundaries can affect the dose response of an output (both locally, and in terms of its average). If only the input signal and the output average are measured, and the localisation is not accounted for, a mere change of location for the localised node might be inferred as a change in the kinetics/total concentrations of species associated with signal transduction (Supplementary Fig. 5). Such pitfalls can be strongly accentuated in more complex networks, where basic effects of localisation can combine with other features of the network in a way which is essentially obscured by partial measurements.

We now examine cases of network nodes with species of different diffusivities. Nodes with different diffusivities for active/inactive forms are a key feature of multiple intracellular pathways. A key ingredient in intracellular spatio-temporal organisation is interconversion between slow and fast diffusing forms of a protein: for example, complexes with different cytoplasmic mobilities in *C.elegans* zygote[24,25], and cytosolic and membrane bound forms such as the Min system in *E.coli* and polarisation circuit in *S.cerevisiae*[21]. Furthermore, feedback regulation of the

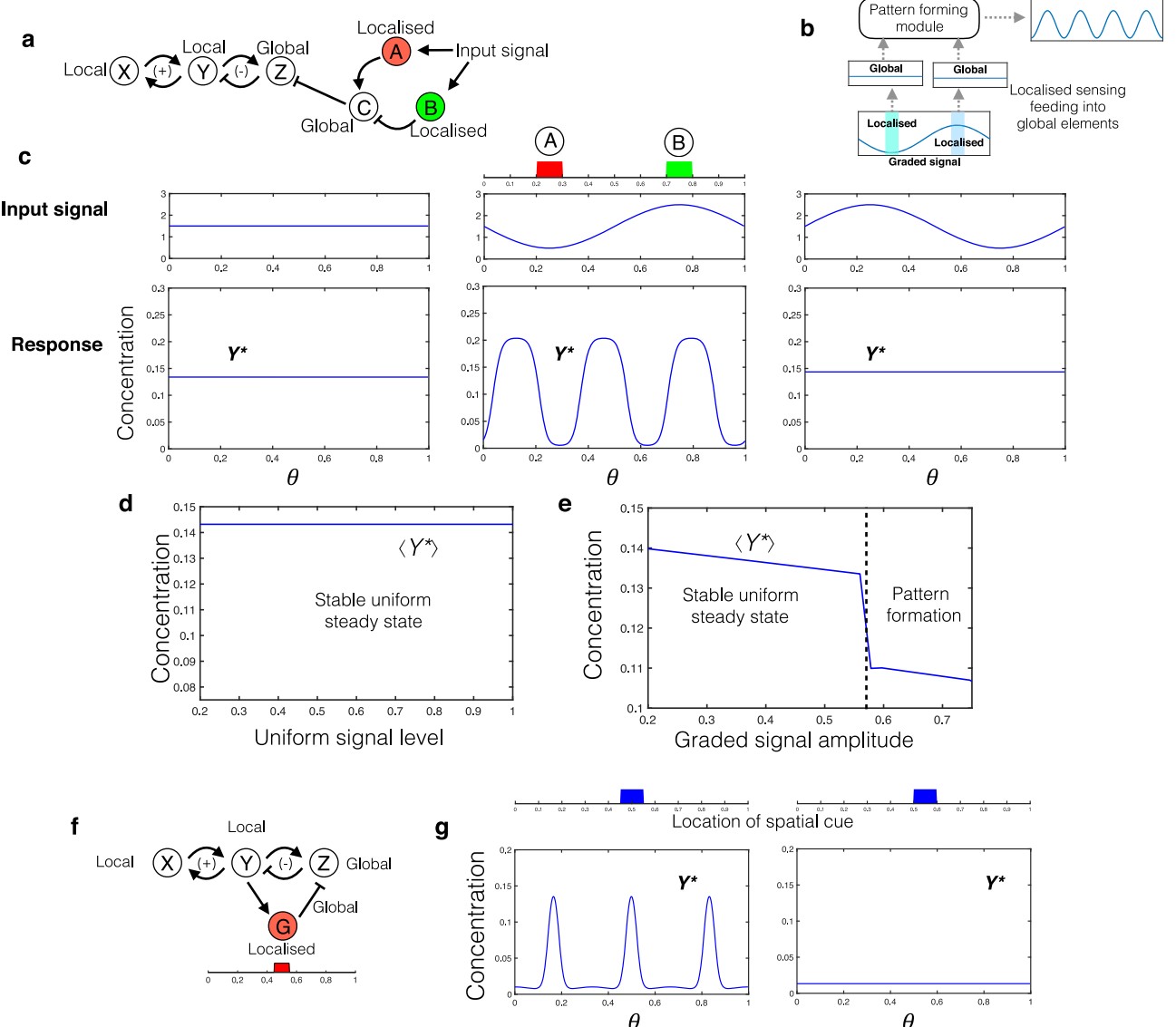

**Fig. 5 Localisation as a regulator of pattern formation. a** Schematic of hybrid pattern-forming motif, where information from the input is processed through two localised nodes feeding into a global node. **b** Schematic depiction of how motif in (**a**) can give rise to patterns in a gradient, without distortions introduced by the gradient. **c** Response to uniform and graded signals. A uniform signal cannot trigger pattern formation, while a graded signal with the same spatial average can produce a pattern. Furthermore, triggering pattern formation also depends on the phase of the gradient in the signal. **d** Varying a spatially uniform signal over a range does not elicit a patterning response: the steady states are uniform with zero spatial amplitude. **e** Spatial average output of node Y as a function of the amplitude of the graded input signal (signal average same as in (**c**)). The discontinuity indicates the onset of pattern formation, as the amplitude of the gradient is increased. **f** Schematic of a feedback control structure allowing for phase selection in pattern formation (periodic boundary conditions). **g** The initial conditions are set by a transient spatial cue (input to Y). Initial conditions with a peak in Y at (or close to) the location of the localised feedback node G lead to a steady state pattern (left panel) with a peak in Y at this location, while other initial conditions do not (right panel). This demonstrates how patterns with particular phases may be selected for, by the confluence of localised and global elements in feedback.

interconversion between forms with different translocation rates between nucleus/cytoplasm is shown to be a key ingredient in an interphase to mitosis switch in *S.cerevisiae*[4]. In such cases, the local concentration of the output of a node at different locations within the domain can exhibit qualitatively different dose response behaviour—increasing in one region and decreasing in another. We demonstrate this in the case of a single node with different diffusivity for the active and inactive forms, with a localised inhibitory signal (Fig. 6d, e), also see Supplementary Note 4 for analytical work). It also follows from this that it is possible for local and average measurements to exhibit qualitatively different trends (Fig. 6e, f, g). These qualitatively different

trends can result in qualitatively incorrect inferences about interactions between nodes, as can be easily seen in simple networks. The fact that ignoring spatial organisation can lead to incorrect causations from observed correlations is highlighted by these simple networks.

We now discuss how incorrect network structures may be inferred in such cases. Basic feedback and feedforward motifs are associated with certain qualitative behaviour based on their network structure—for instance positive feedback with bistability, negative feedback with oscillations etc. Here we show that, if we aim to consolidate the observed behaviour (as obtained from data) in terms of a purely temporal network picture, the network

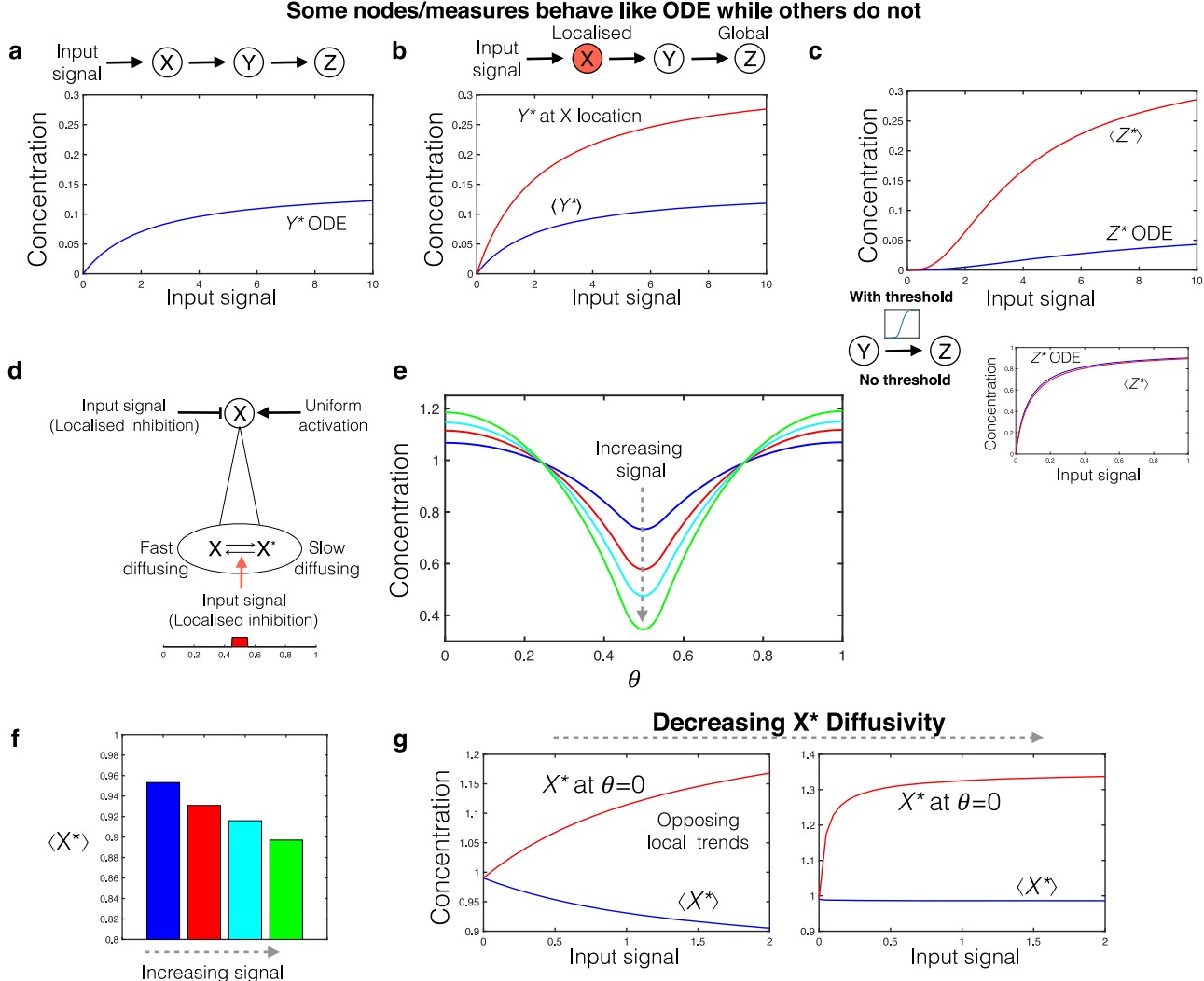

**Fig. 6 Measurements and spatial organisation. a** Simple signal transduction cascade, no localisation: dose response of node Y (identical to ODE model). **b** Localising the input node X (fixed total amount): while node Y exhibits gradients, its spatial average may still exhibit a quantitatively similar dose response to the ODE model seen in (**a**). However, there is a significant difference between local (at location where X is localised, red curve) and spatial averaged measures of output (blue curve). **c** In this case (spatial average measurement), if the regulation of node Z by Y involves no threshold behaviour, the dose response of node Z (spatial average) may be similar, even quantitatively, to the ODE model. If it does involve a threshold, even the spatial average dose response (red curve) can be qualitatively different to the ODE model (blue curve). **d** Schematic of a node with slow diffusing active form and fast diffusing inactive form. Input signal consists of a localised inhibition at the centre of the domain. **e** Contrasting responses to input within and away from the input location: decreasing at the input location (consistent with increasing inhibition) and increasing away from this location. **f** The domain average response exhibits a (decreasing) trend consistent with increasing inhibition. **g** Contrasting dose response in terms of local concentration of $X^*$ (red curve) and global average of $X^*$ (blue curve). For even lower diffusivity of $X^*$, the change in average response to increasing signal becomes negligible, while the local level of $X^*$ exhibits a prominent 'inverse' response.

so obtained may be significantly different from the actual interaction motif. To do this we examine a two-node motif with a feedback loop, with one of the nodes (X) having a low diffusing active form and a high diffusing inactive form (Fig. 7), reminiscent of classes of circuits encountered biologically. X activates the node Y, which is global, and Y in turn inhibits node X. Thus, in the absence of any localisation, the two nodes form (and behave like) a negative feedback loop (Fig. 7a–c). However, on localising a particular interaction—the inhibition of X by Y, we see that the motif essentially behaves like a positive feedback loop, even producing bistable characteristics (Fig. 7d, e). Reconciling such characteristics within a temporal network, could result in a completely incorrect inference about the nature of the feedback (also see Supplementary Notes 4.3, 4.4). Fig. 7f shows dose response curves for X at different spatial locations,

and also demonstrates that spatial averaging can mask the presence of multistability.

To generalise the above result, we consider two-node feedback circuits where one node is a two diffusivity node. We find (see Fig. 7g–i), Supplementary Note 4.5): (i) If the slow diffusing form is the active form, then localisation of the two interactions in different spatial regions can result in 'deviant' feedback behaviour (negative feedback behaving as positive feedback, as seen previously, or vice-versa); co-localisation of both interactions ensures this will not happen. (ii) Suppose both forms of the two diffusivity node have activity. If the fast diffusing form is the more active form, then co-localisation of interactions can result in 'deviant' feedback behaviour, while localisation in different regions prevents it. (iii) In the above instances, even when the feedback behaviour is maintained,

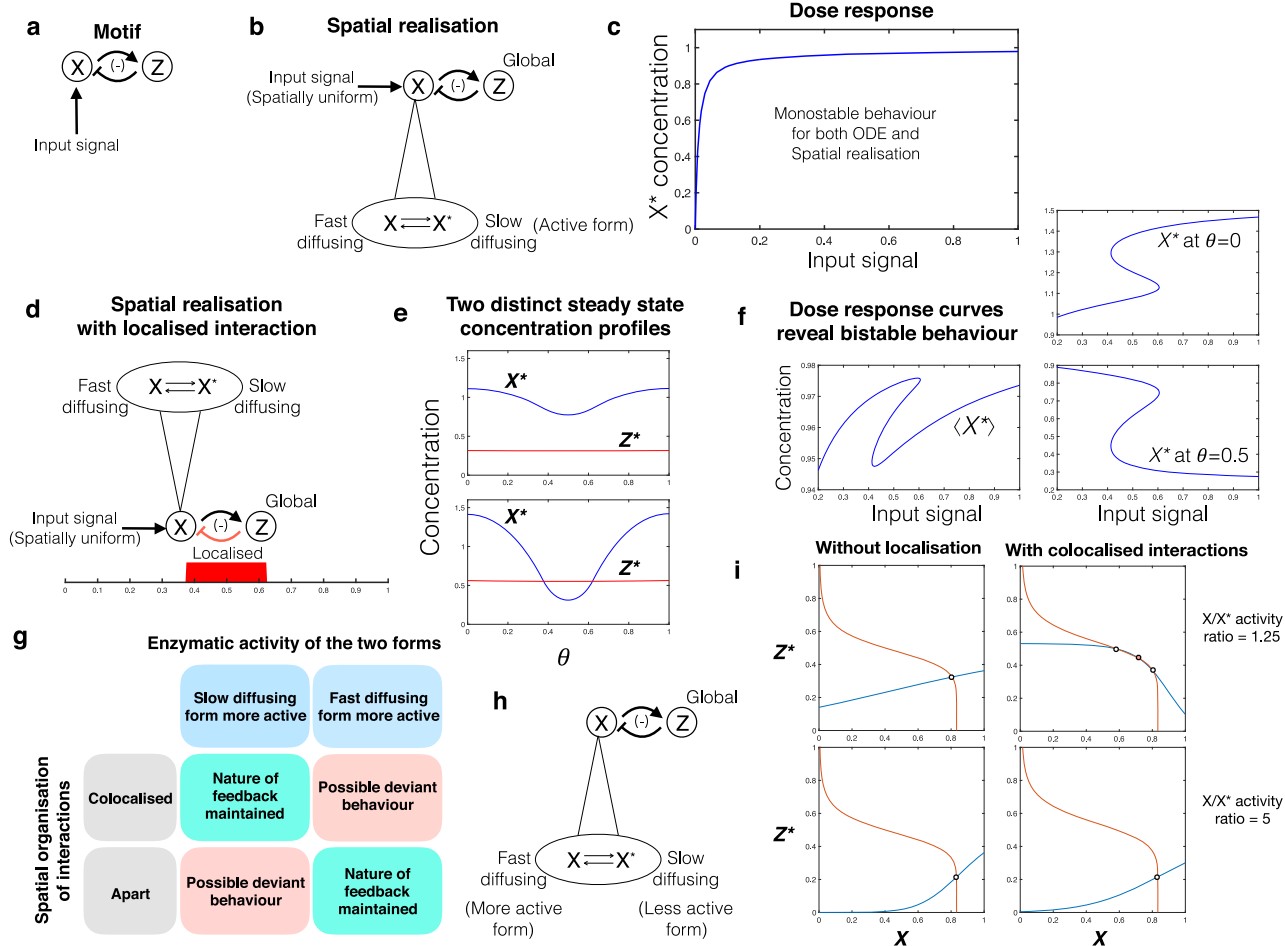

**Fig. 7 Inferring the nature of a feedback motif. a** Schematic of the negative feedback motif (purely temporal realisation). **b** Schematic of the spatial realisation with the X node involving interconverting forms with different diffusivities. **c** Bifurcation analysis of the ODE model shows monostable behaviour. The corresponding PDE model exhibits bifurcation characteristics identical to that of the ODE model, for a uniform input signal. **d** Schematic indicating localisation of one of the interactions: inhibition of X by Z. **e** This system can exhibit bistability. The two co-existing stable steady state profiles for a fixed input level are shown here ($X^*$ blue curves, $Z^*$ red curves). **f** Bifurcation analysis of the PDE model reveals bistable behaviour. However, if the domain average is used to characterise the steady state(s), the contrast between the steady states may not be significant, and a transition from one branch to the other may be consequently obscured (left panel). The same steady state can be characterised as a high $X^*$ or a low $X^*$ if a local measure of the output is used to characterise the steady state (two right panels). **g**–**i** Characterising feedback behaviour in circuits with two diffusivity nodes and localisation of interactions. **g** Assessing the impact of different types of localisation in a feedback circuit containing a two diffusivity node. The type of circuit is depicted in (**b**). This is depicted in the table which is organised along two axes-the localisation of the interactions (whether in the same location or different locations) and the relative activity of the slow and fast diffusing forms. The table is relevant to both negative and positive feedback and highlights scenarios which can lead to deviant feedback behaviour (see text). **h** depicts the specific instance of a negative feedback circuit where the fast diffusing form is more active (but both forms are active). **i** Nullcline-based analysis (see Supplementary Information) shows that when interactions are co-localised, in certain ranges of activity ratios, this feedback circuit can exhibit bistability (crossing of nullclines at three points: red curve X-nullcline, blue curve Z nullcline).

---

there are spatial regions where the feedback can have anomalous effects.

### Relevance to engineering biomolecular systems: homoeostasis and circuit rewiring

*Adaptation and Homeostasis.* Control for adaptation and homoeostasis is a fundamental aspect of biomolecular networks. When spatial organisation is combined with the network, this leads to two complementary questions: (1) how does spatial organisation (localisation) affect the homoeostatic capabilities of the network? (2) can networks be designed to enable homoeostasis to spatial factors such as distance? We discuss both these aspects.

How does localisation impact homoeostasis/adaptation? We study this in two canonical circuits, the negative feedback and

incoherent feedforward motifs[26]. (1) If the output node is not highly diffusible, then introducing localisation (in either one of the feedforward arms or feedback) results in an erosion of homoeostatic capabilities even to spatially uniform stimuli, as a consequence of gradients in the output (even if the spatial average of output node is the output of interest: see Supplementary Information, Supplementary Fig. 8). In this regard, we point out that if the output node was an open node (i.e. with production and removal as opposed to interconversion), then the spatial average of the output adapts to a homogeneous stimulus, even if the output profile does not. This further reinforces how particular measurements can mask key aspects of underlying behaviour. We then turn to the case of a highly diffusible output node where steady state gradients are negligible and adaptive behaviour to uniform stimuli is maintained. In such a scenario we find that: (2)

localising the negative feedback node (Fig. 8a) essentially allows for the maintenance of the same homoeostatic capability (as the non-localised case) provided the total amount of species in this node is maintained. Adaptive responses to both homogeneous and graded input signals ensue. (3) By contrast, localising one of the feedforward pathways (Fig. 8b) in an analogous way essentially distorts the adaptive response resulting in non-adaptive responses to gradients. This distortion arises as a

consequence of two distinct types of information processed through the arms of the feedforward circuit, regulating the (global) output: information about the entire stimulus (local) through one arm, and information about the stimulus at a particular location (localised), through the other. These generically do not cancel out. This represents a distinct type of non-adaptive spatial computation in a gradient consistent with adaptation in a homogeneous stimulus. (4) We show that by

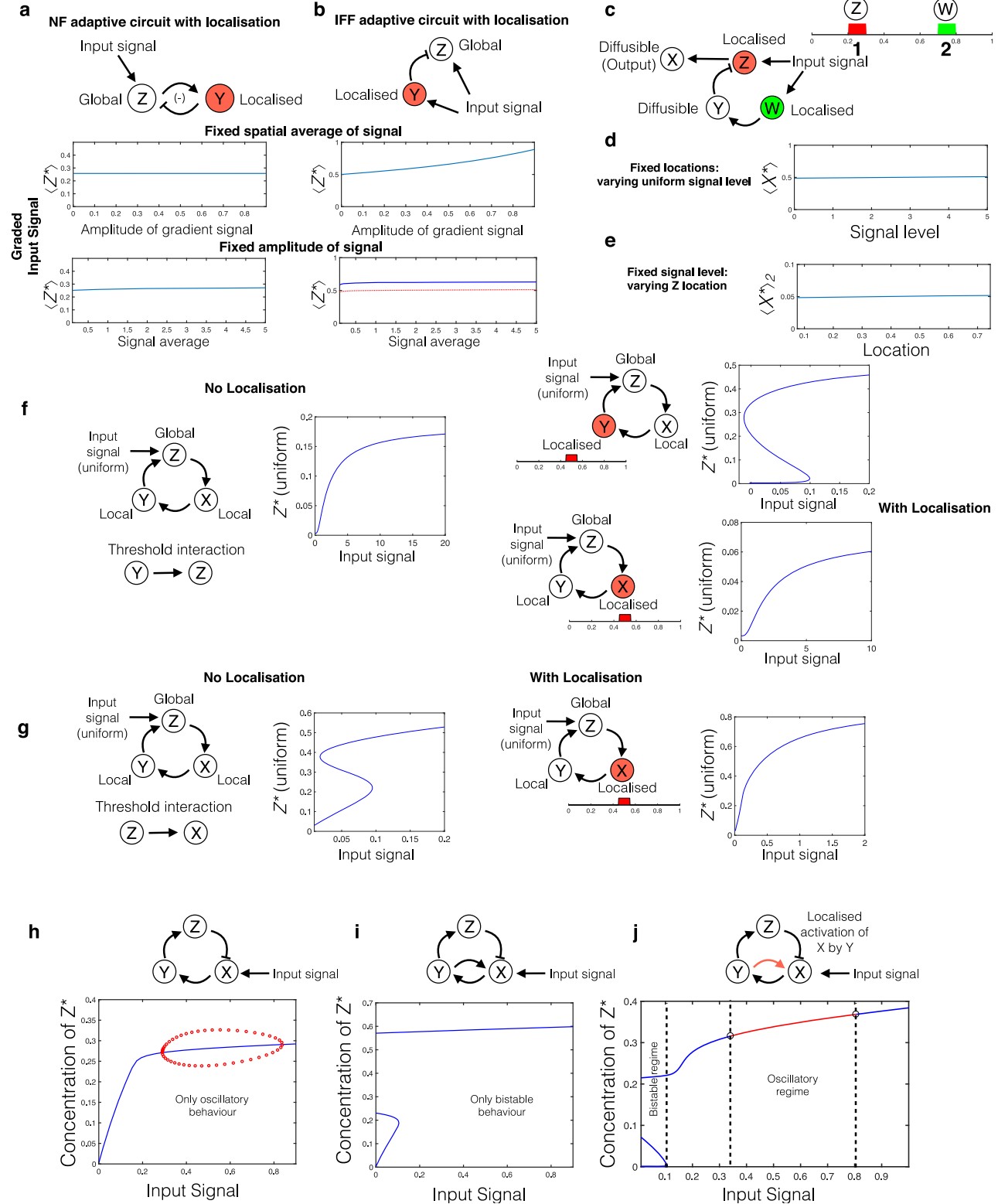

**Fig. 8 Localisation and engineering: applications to homoeostasis and circuit rewiring. a–c** Control structures. **a** Imposing localisation on a homoeostatic negative feedback motif with diffusible (global) output: capable of adaptation to both homogeneous and graded input signals. **b** Imposing localisation on one arm of an incoherent feedforward homoeostatic motif with diffusible (global) output: capable of adaptation to homogeneous signals, but not to graded signals. However, for a fixed amplitude of signal the output adapts to changing signal average (red dotted curve: uniform signal, blue solid curve: graded signal). **c** Motif capable of combined adaptation to signal and spatial separation. **d** For fixed locations of nodes Z and W (compartments 1 and 2), the output average across the domain (and the output profile itself) essentially adapts to changes in a homogeneous signal level. **e** For fixed homogeneous signal level, the output level at compartment 2 essentially adapts to changing location of compartment 1. **f–j** Spatial rewiring to circumvent kinetic constraints. **f** With the Y (local) to Z (global) interaction involving threshold behaviour, localisation of node Y can enable the motif to exhibit bistability. On the other hand, localising the other local node (X) does not achieve the desired effect (see text). **g** In contrast, with the Z (global) to X interaction involving threshold behaviour, localisation of node X can work in the opposite direction. In fact it can weaken/destroy bistable behaviour, if it was present initially. **h–j** Modular combination of characteristics. All nodes moderately diffusible. **h** Negative feedback motif giving oscillatory behaviour. **i** Introducing positive feedback interaction between nodes X and Y produces bistability, but destroys oscillatory behaviour. **j** Both behaviours become accessible when the positive feedback interaction is localised.

combining localised and global elements in each leg of an incoherent feedforward loop yields a distinct "gradient detector" consistent with adaptation in homogeneous stimuli, without requiring the output node to be highly diffusible (and without introducing output gradients). This works by pitting responses to signal levels at prescribed locations against one another to determine the response (see Supplementary Note 5).

Is it possible to buffer against localisation? We now focus on the complementary problem of which control structures can buffer against spatial factors. We demonstrate how it is possible to buffer against the changes in distance between two locations (also see[27] in the context of genetic circuits). To do this we consider localised activation of a moderately diffusing output node and focus on the output level at a fixed (different) target location. This output level can be made to adapt to changes in separation by introducing negative regulation of the output. This regulation is performed by a diffusing inhibitor that is activated (independent of the input signal) at the target location. Analysis (see Supplementary Notes 5.1, 5.2) indicates the confluence of factors responsible for this: (i) equal ratio of diffusivity to reverse conversion rates for the output species and inhibitor (ii) the inhibitor must target the input node, rather than inhibit output directly (iii) All the nodes are far from saturation (a significant fraction of inactive species).

This mechanism can be incorporated into larger motifs, enabling complex homoeostatic tasks such as adaptation to distance as well as input signal. Having a signal regulate both the input node and the (localised) activation of the diffusible inhibitor above allows for output at target location to be independent of signal as well as distance (see Fig. 8c–e), Supplementary Note 5.3) This represents the confluence of two strands studied above: the imposing of spatial organisation on conventional (temporal) adaptive motifs, and the creation of motifs dealing specifically with adaptation to a spatial factor.

The basic ingredients considered above: localised activation/ production, global repression, and changing spatial parameters, form key ingredients of homoeostatic mechanisms in developmental systems, such as the expansion-repression mechanism for scale-invariant patterning in growing domains[28,29]. Our analysis allows us to examine how buffering against size/distance may be implemented at both intracellular and tissue levels.

*Circuit/Network Rewiring.* Rewiring of networks is an experimental tool used to engineer biomolecular networks[30]. We now turn to a distinct aspect of spatial organisation: spatial 'rewiring' to circumvent kinetic constraints.

Spatial rewiring can enable modular combination of circuits, as we now demonstrate. We consider two circuits which share common elements: the behaviour of these circuits in isolation are characterised: one a negative feedback circuit generating

oscillations, while the other circuit, sharing the same backbone yields bistability (Fig. 8h–j). However, the combination of both circuits results in the abrogation of oscillation and only bistable behaviour being realised, for the basal parameters. By employing localisation (in this instance) of the element involved in the positive feedback loop, and varying the extent of localisation, it is possible to obtain a range of behaviour, which includes (1) the basal kinetic behaviour—bistable (requiring no localisation) (2) oscillatory behaviour without bistability (for sufficiently small localisation of the positive feedback element) (3) a combination of bistability and oscillations, which could be regarded as a desirable combination of the behaviour of the two circuits (Fig. 8j). For a fixed diffusivity (possibly high), tuning the size of localisation can result in the desired behaviour (demonstrated in Supplementary Note 6).

Localisation of the positive feedback element and the negative feedback element in different locations affords further possibilities. To start with, it could circumvent any undesired interaction between these elements should they exist. Furthermore, this opens up multiple avenues for the modular combination of these circuits. In one instance, if the common elements are non-diffusible/weakly diffusible, this results in essentially, decoupled circuits at different locations which can be coupled by a global downstream element. In this manner, we circumvent the constraint caused by common elements. Direct coupling without requiring downstream elements, can also be obtained for weak to moderate diffusivity of common elements (see Supplementary Information).

Localisation can also enable circuit design by alleviating kinetic constraints through local elevation. The next example highlights both the use of localisation as well as a design perspective to assessing its effect in networks, something crucial to its effective use as circuit design/rewiring tool. The capacity of localisation to alleviate kinetic constraints is demonstrated by it enabling bistability in a two-node positive feedback circuit, with a threshold interaction, where bistability was initially absent. If the circuit originally involved only local elements, such localisation could enable bistability but would constrain bistability to the particular region of localisation. If the original circuit contained a global element, this is no longer an issue and localisation has the desired effect if (i) the threshold occurred in the local-to-global interaction and (ii) localisation is imposed on the local element: localisation causes concentration elevation enabling crossing of thresholds. This would not work if the threshold was in the global-to-local interaction as localisation has a purely neutral effect (see Supplementary Note 6.1). In summary localising the regulating (local) node in the threshold interaction can have the desired effect (see Supplementary Information).

Now we consider a slightly more complex version of the previous case where there are multiple local nodes which are

candidates for localisation (Fig. 8f, g) depicts the case with two local nodes and one global node). The requirements for localisation to enable bistability are that localisation is applied to the element immediately preceding the threshold interaction and that it applies to a local node (this implies that the threshold cannot be in the global-to-local interaction, as localising local elements in this instance has an essentially neutral effect). In such a circuit (involving a threshold interaction in the local-to-global interaction), there could be multiple local nodes, 'upstream' of the threshold interaction, and candidates for localisation. A new emerging insight is that localisation of an element not immediately preceding the threshold interaction can in fact move the system further away from bistability. This is because: (i) this can result in a tradeoff with a local increase of the downstream node at the location and a reduction elsewhere, (ii) saturation may limit the impact of the local increase, with the decrease elsewhere dominating (see Supplementary Information, Supplementary Fig. 8).

**Exemplar cases**. We illustrate multiple themes of the paper by exploring two concrete biological contexts, involving localisation and circuits with two diffusivity nodes. A common feature is the key role of localisation of Polo-kinase and regulation of the associated circuits. The results are concisely summarised here.

By dissecting the three-tier cascade underlying gradient transduction and polarisation in *C. Elegans* (see Methods, Fig. 9b, d–f), Supplementary Note 7) we find:

(a) Analysis of cascade building blocks (single tier: two diffusivity node): analyzing localisation of interconversion of species in both directions reveals contrasting features for gradient generation, in particular the possibility of a non-monotonic input dependence if localisation regulates conversion of the fast diffusing form. (b) Impact on cascade behaviour: by examining two different realisations of the three-tier cascade (maintaining the same input–output relationship), we find significant differences both when (i) a threshold is present in the interaction and (ii) there is a mis-localisation of a component enzyme in the intermediate tier (involving Polo-kinase). (c) Implications: This reveals underlying design principles/constraints for gradient transduction, and the impact of localisation. It suggests that the biological observed design has an inherent advantage for realising the same outcome, bypassing such non-monotonic effects, and is consequently more robust.

By evaluating and analyzing postulated circuits underlying spatial cell-cycle switches in *Drosophila* (involving a positive feedback circuit, along with a negative feedback: see Methods, Fig. 9c, g–l), Supplementary Note 7), we find (a) Spatial switching: the positive feedback circuit with two two-diffusivity nodes enables spatial switching, of the type seen experimentally. (b) Negative feedback: the negative feedback interaction modulates spatial switching thresholds and amplitude. (c) Deviant feedback behaviour: Fig. 7 reveals ways in which each type of feedback can exhibit an opposite type of behaviour, something observed when an enzyme is mislocalized (Fig. 9 (l)). The negative feedback studied can also exhibit anomalous behaviour in the nucleus if both forms of Polo have activity. (d) Relative size dependence: changing the relative size of the two spatial domains (nucleus, cytoplasm) can significantly impact both feedbacks and system behaviour, and also facilitate anomalous/deviant feedback behaviour.

## Discussion

The widespread features of biomolecular networks and spatial organisation, and the diverse nature of their interplay pose many fundamental questions. Addressing these questions requires a theoretical framework transcending the individual context. The emerging insights are foundational to natural and engineered

biology. Rapidly emerging techniques for spatial measurement (imaging, spatial proteomics)[31], engineering spatial organisation (in bottom-up synthetic biology), and the continual blurring of boundaries between natural and engineered biology provides further impetus for such a study (Fig. 10).

**The effect of spatial localisation**. Spatial localisation can impact network behaviour in multiple non-intuitive ways, fundamentally altering it, creating capabilities which cannot be seen in purely temporal (lumped) network analogues (Fig. 10).

Localisation is used by evolution to enable specific capabilities, such as eliciting different signalling outcomes by different ligands (in calcium signalling[32], enabling intergenerational memory (pheromone response of a cell-cycle switch[5] and spatial switching[4]; mislocalization of key components is implicated in dysfunction and disease[33] The interplay of localisation, diffusivity and size is central to designing cell-free synthetic pattern-detection circuits, and in tissue engineering[14]. Localisation is an essential tool in the construction of chemical sensing materials using modular synthetic circuits[34], building collective information-processing systems, combining communicating cells (natural/artificial) containing different building block circuits[13,35]. Dynamic control of localisation is an emerging tool, both at the intracellular level—through optical control and synthetic scaffolds, and at the population-level through control of cell adhesion[36–38]. Our dissection of the impact of localisation amidst different network structures, kinetics, functionalities provides a basis for mapping out essential transitions arising from alteration of localisation thus delineating the new capabilities it provides. In our studies, the interaction motif continued to appropriately describe interaction between components. Localisation when occurring in multiple locations can also result in a change in the interaction network structure by eliminating interactions between entities not in the same physical location, something which could significantly impact network behaviour.

**Localisation and the complexity of networks**. Systems and synthetic biology span different approaches, depending on the problem being addressed. Systems biology studies at the intracellular level range from study of fairly small scale networks (key drivers of behaviour investigated) to networks of moderate complexity to large scale studies with a fairly large number of species being involved. Existing studies focussed on spatial aspects, either at the intracellular or intercellular tend to involve small scale or moderate networks. Synthetic biology studies focusses on bottom-up approaches for design of circuits in cell-free or cellular systems, or the rewiring of existing networks. We first note that due to advances in imaging and spatial proteomics, along with manipulation of spatial localisation, the location of many species can be tracked/manipulated. For studies of small (ish) networks, our approach bringing the interplay of localisation and networks is directly relevant: it shows how the key behaviour may be altered, how different networks may give rise to a given behaviour etc. With moderate networks (such as our exemplar studies), again the approach can be used appropriately, by (i) identifying the key drivers of behaviour within the network and (ii) assessing where localisation is present and where it may affect network behaviour. When the network becomes large, then the approach depends on whether spatial data is available or not. If it is, then the data can be the basis of a model which incorporates location as well as network interaction. Here, in addition to specific insights, our approach of dissecting the network ('network centric' and 'space centric') will be helpful in assessing system behaviour. If however, a large scale network is described in ODE terms, a reasonable question to address would be whether

spatial localisation alters essential behaviour. Here (i) accounting of location may reveal that some network interactions may be absent and result in an altered network. (ii) key core pieces of the network could be described both with and without localisation to elucidate key differences. Assessing the role of localisation in larger networks presents many challenges (i) Both analysis and inference may be computationally challenging (ii) There is a greater complexity and a range of possibilities (iii) it will be

necessary to assess and discriminate not only between different possible network interactions, but also different spatial architectures. For the latter, focussed experiments involving the manipulation of localisation, along with spatial measurement may be especially relevant. It should also be emphasised that many key consequences of localisation can be studied effectively by examining these in smaller subnetworks, as it is likely that the key impact of localisation will already be found there. Space-centric

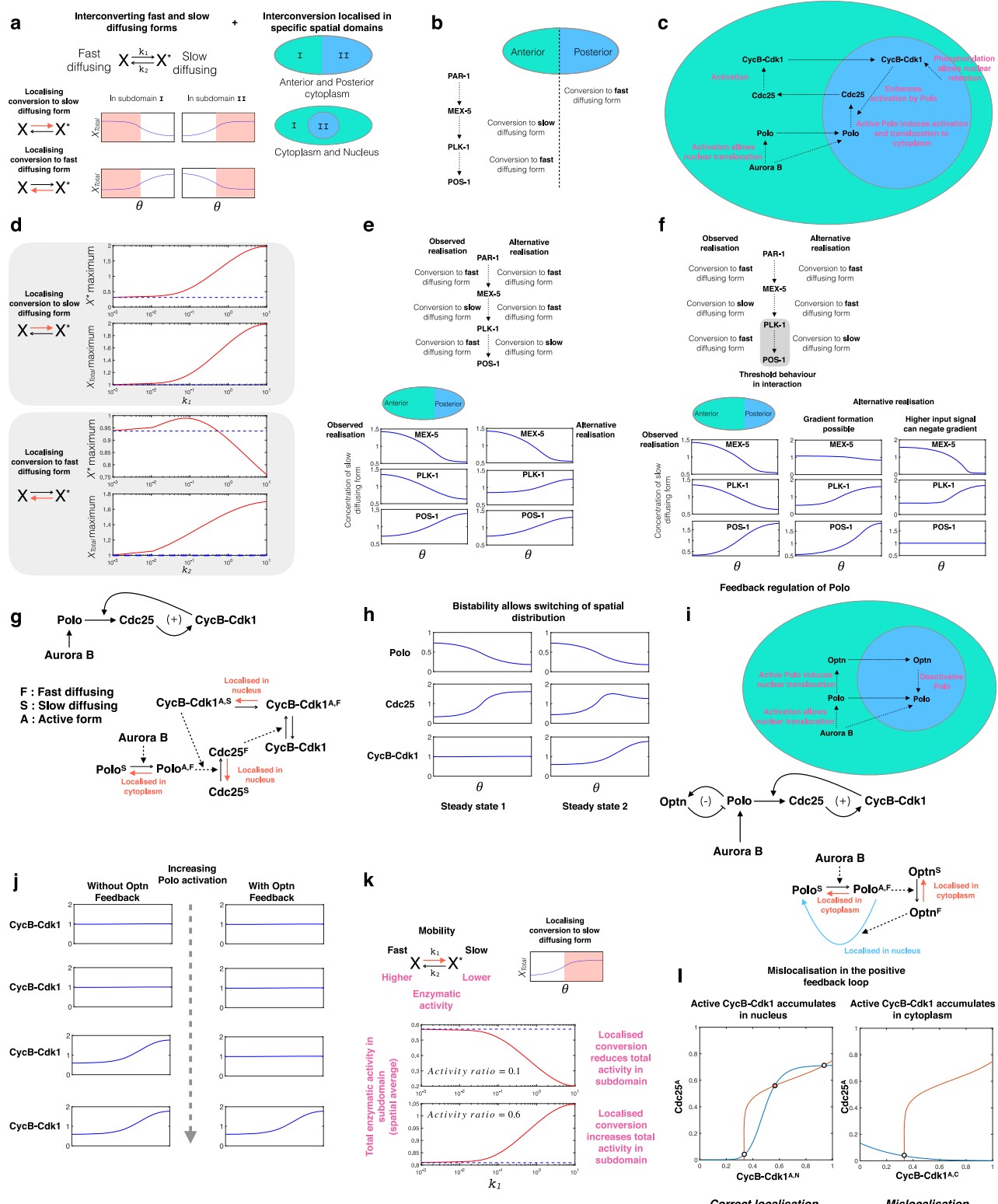

**Fig. 9 Exemplar cases: polarisation and cell-cyle regulation and the role of Polo regulation therein. a** A basic building block of circuits considered here is the two diffusivity node: the results of the two diffusivity node in isolation, as well as different ways in which localisation may be imposed are depicted. **b, c** Intracellular contexts involving Polo-kinase based circuits, along with localisation. **b** A cascade responsible for polarisation in the *C. elegans* zygote. **c** Cell-cycle circuit postulated in *Drosophila*. **d–f** Polarisation in *C.elegans*. **d** Imposition of localisation of different steps in the basic building block. Localising conversion to the fast diffusing form can result in non-monotonic input–output response. **e** Two different realisations of the cascade with similar input–output responses, but contrasting distributions of the Polo kinase. **f** Introducing threshold dependence of POS-1 by Polo Kinase reveals striking contrasts between the two cases. **g–l** Postulated cell-cycle circuit in *Drosophila*. **g** A schematic of the underlying circuit, followed by a depiction of how localisation is imposed. **h** The spatially distributed circuit involving CycB-Cdk1 and Cdc25, can exhibit bistability, which serves as the basis for the mitotic switch. **i** A schematic of a negative feedback regulation of Polo via Optn: this is overlaid on the Cdc25-CycB-Cdk1 circuit considered previously. **j** A depiction of how the Optn feedback affects switching thresholds: switching (indicated by a transition in the Cdc25 profile, from a strong gradient to a weak gradient) requires higher Polo activation. **k, l** Design principles, constraints and anomalies associated with feedback. **k** Anomalous outcome in specific locations: localised deactivation, as seen in the case of Optn regulation of Polo, can in fact enhance the activity of the target, rather than reduce it. This depends on the activity ratio of the two forms. **l** Deviant feedback behaviour: accumulation of active CycB-Cdk1 complex in the cytoplasm, possibly resulting from the mislocalization of an enzyme, can cause the positive feedback circuit to behave like a negative feedback circuit (see Supplementary Information). Red and blue curves denote the $Cdk1_{A,N}$ and $Cdc25_A$ nullclines respectively.

approaches can create a basic template structure within which networks can be organised and studied, building from moderate size networks, and gradually increasing network complexity.

Our approach directly lends itself to both studies in systems biology and bottom-up approaches in synthetic biology, where spatial localisation is a focal point. In rewiring networks in synthetic biology, analysis using models of modest complexity could drive design approaches which could be tested and evaluated on real networks to check if the desired outcome is obtained.

**Network inference**. Reverse engineering of networks is commonplace in biology, involving developing network models based on data, often neglecting spatial aspects. Non-spatial data is used to justify the development of purely temporal network models, either explicitly or implicitly. However our analysis shows how (i) the result can depend crucially on the type of measurement, even if non-spatial (ii) trying to encompass spatial networks into a non-spatial framework is associated with fundamental limitations, and ad-hoc fixes, can lead to erroneous conclusions about other aspects of the network (iii) Some network nodes (for a given type of measurement) may behave like the ODE, while others do not do so, even qualitatively (iv) In employing spatial measurements, the choice of nodes to be measured critically depends on how localisation enters the network-guidelines from lumped analogues of such networks are fundamentally misleading.

We found that with networks containing nodes with different diffusivities, qualitatively incorrect inferences emerge in very simple cases. There are also multiple instances in networks more generally (by ignoring localisation) where an inference of an interaction between two species may be made, where none exists (see Supplementary Note 4.3 for details). As examples, suppose two components are in different spatial compartments and non-interacting. Working in an ODE framework could result in (i) incorrect inference about their interaction based on in-vitro data or data in another context (ii) A correlated regulation of these two components (eg global regulation) could be misinterpreted as an interaction (iii) The possibility of a common factor functioning in different ways in two compartments (enabled by localisation, absent otherwise) and regulating these may be ignored, leading to a misinterpretation of interaction between the two components (for e.g. bifunctional enzymes as discussed in[39] but this applies more broadly). Furthermore, working in an ODE framework can lead to incorrect inferences about the nature of some interactions, which can then directly lead to other postulated interactions (non-existent) inferred, to explain network behaviour. Finally, one node affecting another node's localisation may be inferred incorrectly as an interaction.

Our analysis suggests ways of alleviating problems of incorrect inference (multiple measurements, spatial measurements, systematically reducing spatial models to temporal models where possible).

**Localisation, pattern formation and engineering information processing**. Localisation combined with global components can generate pattern formation which (i) takes its cue from localised upstream/environmental signals (ii) has no distortion due to localisation. This provides a template for constructing classes of hybrid pattern-forming systems comprising classic pattern-forming systems and environment-driven spatial "patterns". This is directly relevant to elucidating hierarchical pattern generation in developmental biology (with intricate combinations of pre-patterning and self-organising elements, which may be difficult to take apart[22]; this brings a new dimension to engineering and controlling pattern formation for tissue engineering. Our results highlight localisation as a generator of pattern formation, potentially making it an easy to manipulate but potent tunable dial.

The different dimensions of localisation in homoeostasis–whether as a distorter of homeostaic behaviour, or a focal point around which homoeostatic/adaptive circuits need to be designed (buffering against location/distance) are relevant in natural and engineered biology. Localisation facilitates modular combination of circuits, eliminating undesirable interactions and overcoming kinetic constraints, and provides a new dimension to circuit rewiring making it potentially a key enabling tool in synthetic biology.

Our study reveals multiple systems facets of localisation/spatial organisation in biomolecular networks both when space is the focal point and when it is not, Multiple insights, arising as they do from the interplay of local, global and localised components in networks, may also be relevant well beyond the biological realm, in areas ranging from physics to engineering to social science.

## Methods
In order to be able to identify clear patterns in the interplay between space and network structure, we consider three limiting cases for the spatial characteristics of components, forming three essentially distinct spatial classes: (i) local components: these components are present everywhere in the spatial domain and are essentially non-diffusible (or weakly diffusible), (ii) global components: these components are highly diffusible across the whole domain, leading to essentially homogeneous steady state profiles and (iii) localised components: these components are non-diffusible and confined to one or more sub-domains. In specific cases, where a clear trend may be discerned, we will also consider the effect of varying the diffusivity of global or local components to make focussed points.

The nodes/components in a network motif, and the interactions between them, may be realised in different ways. In a signalling pathway for example, possible realisations include a node consisting of a pair of interconverting modified forms (with other nodes possibly regulating the forward and/or reverse modification reactions) or a single species (with its production and/or degradation reactions regulated by other nodes). For the purpose of our study, we will consider nodes consisting of interconverting modified forms (one active and the other inactive),

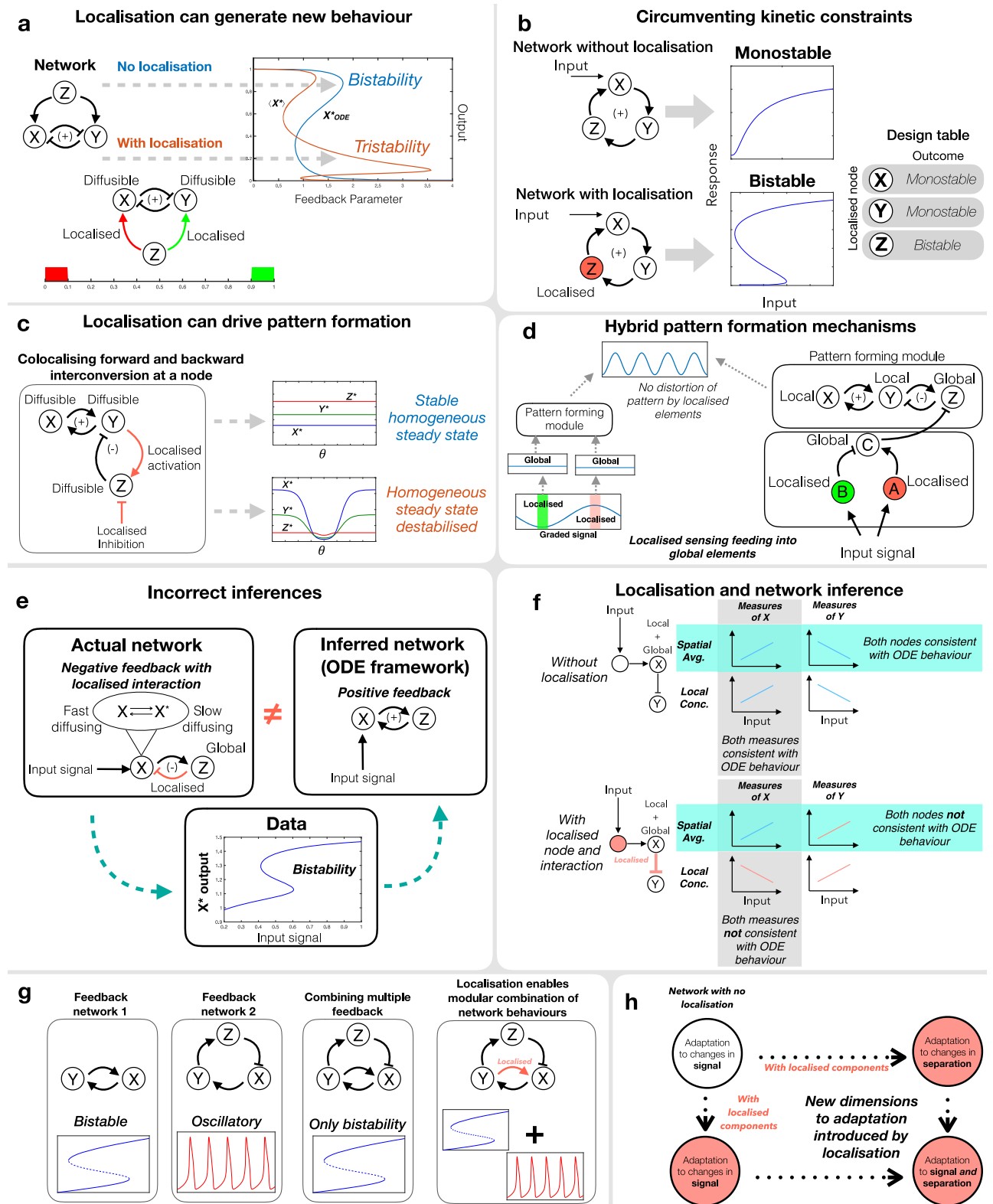

with a conserved total amount at each node, and the interconversion between modified forms regulated by the active form of one or more nodes. The nature of the regulation can be activation or inhibition, both of which can be realised in various different ways. For this study, we restrict ourselves to the following ways of realising activation and inhibition: for activation, the active form of the 'regulating' node converts the inactive form at the 'target' node to the active form; for inhibition, the active form of the 'regulating' node converts the active form at the 'target' node to the inactive form. This type of enzymatic regulation, in principle, involves the formation of enzyme-substrate complexes, and could result in the

sequestration of the regulating species at interacting nodes. It is well known that sequestration effects can have non-trivial consequences for the behaviour of pathways and networks. However, since our focus is on the interplay between regulatory patterns and the spatial characteristics of the interacting components, we will eschew such additional complexity by working in the limit where the enzyme kinetics is such that the amounts of these complexes is negligible. This allows us to clearly discern the most basic aspects of the interplay between network motif and spatial organisation, serving as a platform for investigating the effects of sequestration subsequently.

**Fig. 10 A summary of the effects of localisation on networks. a** Localisation of network components—nodes or interactions, can enable a network to exhibit new dynamical characteristics, otherwise inaccessible. A three-node motif exhibiting bistability with no localisation, can exhibit tristability when two of the interactions are localised in different parts of a spatial domain. **b** Localisation can circumvent kinetic constraints: 'spatial rewiring' of a network by localising a specific node can enable bistability. Localisation of specific nodes allows for this, while others do not (see design table), highlighting the value of a design approach. **c** Localisation of forward and backward regulation of a node allows for a spatially uniform steady state in the network. This can be destabilised giving rise to a spatial pattern, illustrating how localisation may be a potent tuneable dial for pattern formation. **d** Localisation can be deployed to create hybrid pattern-forming systems, combining features of gradient-driven patterns and (Turing mechanism-based) self-organised patterns. Aspects of the gradient are key to obtaining pattern formation, though they do not distort the pattern. **e** The presence of localisation can cause bistability in a negative feedback network to be incorrectly inferred (in an ODE framework) as a positive feedback. **f** The presence of localised components can introduce fundamental limitations in network inference based on ODE frameworks. We consider a network with and without localised components (the latter consistent with the ODE) and focus on nodes and measurements, revealing the underlying source of the disparity. With one of the nodes and one interaction localised (i) for the same type of measurement, some nodes are inconsistent with the ODE. (ii) for the same node, two types of measurements reveal opposite trends. **g** Spatial localisation can be a tool for modular combination of circuits (shown for bistable and oscillatory circuits), where a simple (kinetic) network combination does not work. **h** Localisation presents new dimensions to homoeostasis/adaptation which we explore: how localisation impacts (temporal) homoeostasis, how circuits can be designed for adaption to distance between locations, and how that is combined with adaption to signal.

**Models: Kinetics**. We use models based on mass action kinetics to represent the networks. For the kinetic model of a given motif, we have one equation each for the active and inactive forms associated with each node, describing the time derivative of the concentrations of those forms. The reactions to be accounted for at each node include both interconversion reactions: inactive to active and active to inactive, each of which may be regulated by the active forms of other nodes. In addition, there may be basal rates for the same reactions (independent of other nodes). The model for the regulation of one node (labelled Y) by the other node (labelled X) is given by:

For node X activating node Y

$$\frac{dX}{dt} = -k_x X - k_{sx} SX + k_{-x} X^*$$
$$\frac{dX^*}{dt} = k_x X + k_{sx} SX - k_{-x} X^*$$
$$\frac{dY}{dt} = -k_y Y - k_{xy}(X^*)Y + k_{-y} Y^*$$
$$\frac{dY^*}{dt} = k_y Y + k_{xy}(X^*)Y - k_{-y} Y^*$$

For node X inhibiting node Y

$$\frac{dX}{dt} = -k_x X - k_{sx} SX + k_{-x} X^*$$
$$\frac{dX^*}{dt} = k_x X + k_{sx} SX - k_{-x} X^*$$
$$\frac{dY}{dt} = -k_y Y + k_{-y} Y^* + k_{-xy}(X^*)Y^*$$
$$\frac{dY^*}{dt} = k_y Y - k_{-y} Y^* - k_{-xy}(X^*)Y^*$$

(1)

where $S$ represents the input signal to the system, activating the $X$ node. We use the following notation for the parameters here: $k_i$ and $k_{-i}$ represents the basal rate of conversion to and from the active form, for species $i$, while parameters of the form $k_{ij}$ and $k_{-ij}$ represent activation and inhibition of node $j$ by node $i$ respectively.

In our study, we also examine the effect of regulation via a Hill function. Such a regulation, encountered in multiple systems, may be representative of co-operative or otherwise nonlinear mechanisms in the regulation, which may not be modelled explicitly. With a Hill function, the same interactions described above, take the following form:

For node X activating node Y

$$\frac{dX}{dt} = -k_x X - k_{sx} SX + k_{-x} X^*$$
$$\frac{dX^*}{dt} = k_x X + k_{sx} SX - k_{-x} X^*$$
$$\frac{dY}{dt} = -k_y Y - k_{xy} Y \left( \frac{(X^*)^n}{K_{xy} + (X^*)^n} \right) + k_{-y} Y^*$$
$$\frac{dY^*}{dt} = k_y Y + k_{xy} Y \left( \frac{(X^*)^n}{K_{xy} + (X^*)^n} \right) - k_{-y} Y^*$$

For node X inhibiting node Y

$$\frac{dX}{dt} = -k_x X - k_{sx} SX + k_{-x} X^*$$
$$\frac{dX^*}{dt} = k_x X + k_{sx} SX - k_{-x} X^*$$
$$\frac{dY}{dt} = -k_y Y + k_{-y} Y^* + k_{-xy} \left( \frac{(X^*)^n}{K_{xy} + (X^*)^n} \right) Y^*$$
$$\frac{dY^*}{dt} = k_y Y - k_{-y} Y^* - k_{-xy} \left( \frac{(X^*)^n}{K_{xy} + (X^*)^n} \right) Y^*$$

(2)

We examine functions where the Hill coefficient $n$ may be 2 or 4. Models with these Hill coefficients have been widely used in describing protein interactions and signalling pathways.

**Models: Space**. Studying the behaviour of spatial networks involves expanding models of the kinetics of networks to bring in spatial aspects: this involves the description of (i) the spatial domain, (ii) the localisation or diffusion of species in the spatial domain and (iii) the boundaries. Note that an explicit spatial description in necessary for such a study and approximations such as representing transport effects in kinetic terms (or via delays) is not sufficient. For simplicity, we confine ourselves to examining these reaction systems realised in a 1-D spatial domain. Note however, that the basic aspects of the interplay between spatial organisation and network interaction patterns will carry through to higher dimensions, though additional geometry dependent effects could also play a role. We combine the kinetics and spatial organisation in an explicit spatial description (PDE) consisting of reaction-diffusion equations. We can have local nodes where the species are present everywhere in the domain, with their diffusivity set to a relatively low value (possibly zero); global nodes where the species are present everywhere in the domain, with relatively high diffusivity; localised nodes with species confined to certain sub-domains, with their diffusivity set to zero. Note that, with all three classes of nodes, unless otherwise mentioned the two interconverting forms are assumed to be equally diffusible. This simplifies the model, allowing us to describe only the active forms, by using the total concentration to eliminate the concentration of the inactive form, This is because, the equal diffusivity of the two modified forms implies that, if the total concentration of the two species is initially uniform in space, then it will remain so subsequently. This is also true for species localised in a sub-domain. Throughout the study, for all simulations, we will confine our investigation to initial conditions where the total concentration of modified forms, for any node, is uniform within the spatial domain containing that node (whole domain or localised sub-domain). To illustrate the types of models we use, we consider the example discussed above (node X regulating node Y): the model of a localised node X regulating a node Y (which could be local or global) is given by:

For localised node X activating node Y
Within the localised subdomain where X resides

$$\frac{\partial X}{\partial t} = -k_x X - k_{sx} SX + k_{-x} X^* + D_x \frac{\partial^2 X}{\partial \theta^2}$$
$$\frac{\partial X^*}{\partial t} = k_x X + k_{sx} SX - k_{-x} X^* + D_x \frac{\partial^2 X^*}{\partial \theta^2}$$

with $D_x = 0$

$$\frac{\partial Y}{\partial t} = -k_y Y - k_{xy} Y(X^*) + k_{-y} Y^* + D_y \frac{\partial^2 Y}{\partial \theta^2}$$
$$\frac{\partial Y^*}{\partial t} = k_y Y + k_{xy} Y(X^*) - k_{-y} Y^* + D_y \frac{\partial^2 Y^*}{\partial \theta^2}$$

(3)

Outside the localised subdomain
(X and X* being absent here)

$$\frac{\partial Y}{\partial t} = -k_y Y + k_{-y} Y^* + D_y \frac{\partial^2 Y}{\partial \theta^2}$$
$$\frac{\partial Y^*}{\partial t} = k_y Y - k_{-y} Y^* + D_y \frac{\partial^2 Y^*}{\partial \theta^2}$$

In the above equation $D_y$ depends on the nature of Y (either a local or a global node). In an analogous way, model of local/global node X regulating localised node Y can be written.

We consider spatial domains with both periodic and no-flux boundary conditions.While both boundary conditions give comparable insights in many cases (for reasons which are easy to understand), there are also instances where the nature of the boundary (and consequently boundary condition) play an important role: we identify those as such, during the discussion of the results.

We comment on the case of nodes which contain both local and global sepcies. A variation of the above models is necessary in those cases where we have nodes that contain both local and global elements, i.e. where one of the interconverting forms is fast diffusing and the other is slow diffusing. There are multiple instances of such cases in the biological literature. Here we employ equations describing the dynamics of both forms, with the corresponding diffusivities incorporated. In examining networks with such 'two diffusivity' nodes, we find that if the active form is the highly diffusible form, this behaves in many respects like a global node (in terms of its contribution to network behaviour). We also point out that it is possible to have two diffusivity nodes, where one node is localised (in some sub-domain) and the other is global. In such cases, this requires the conversion reaction of the global form to be localised in this sub-domain.

Note: At any given node participating in an interaction, at least one of the interconversion reactions is regulated either by an input signal or by a different node. If only one of the reactions is regulated in this way (which is typical), then we assume that the reverse reaction occurs wherever the species is present, and is associated with a fixed rate constant which does not vary spatially. Codes for the models used are presented in Supplementary Information (note notation).

**Input signal**. An important aspect of the behaviour arising from a pattern of interactions between nodes, is how a designated output (i.e. the concentration of the active form at a specific node) is regulated by an input signal. In this study, we consider input signals acting through the same type of enzymatic regulation as the interactions between nodes. Furthermore, unless mentioned otherwise, we do not examine dynamically varying input signals, except in special cases. In the spatial context, we can have spatially homogeneous as well as spatially graded concentration profiles for the input signal, each of which can elicit different types of response from a given motif. For specificity, to represent spatially graded signals, we employ a sinusoidal signal profile of wavelength equal to the domain size (this is an analogue of a linear gradient in a domain with periodic boundary conditions). The basic insights from the study do not depend in any essential way on this choice of signal. It is worth pointing out that in certain motifs (e.g. feedforward motif), the node at which the signal acts is naturally determined. In other motifs (e.g. feedback motifs), there are multiple nodes at which the signal can act. This is especially significant in our study, as different nodes can belong to different spatial classes. We have examined this aspect in detail as part of our analysis.

We now discuss parameters. Our focus in this study is to identify broad qualitative trends in the functioning of motifs with spatially organised components. To this end, we choose kinetic parameter values for the motifs such that they exhibit the characteristic dynamic behaviour associated with them in the temporal context (as ODEs). The results we present do not rely on non-generic values for these parameters. In all cases, we see that the behaviour we report is observed over a range of values of these parameters, and thus essentially emerges from the structure of the interactions themselves. This is reinforced by a bifurcation analysis of the ODE. If there is a range of parameters which allows for certain behaviour, we choose a parameter set in the middle of this range, not close to the bifurcation points (which determine the range of parameters). In essence, all we require of the motifs (and their parameters) is that they exhibit the behaviour they were expected to show (based on prior studies of such motifs), and that they have the structure that they do.

The values of spatial parameters—diffusivity, domain size and size (and location) of localised sub-domain, are chosen appropriately for the particular spatial realisation being studied. We first comment on nominal parameter values. We keep the overall domain size fixed, as varying this, can be understood in analogous terms to varying diffusivities. The size of the localised sub-domain is typically one-fifth of the size of the overall domain. The diffusivity of global species is chosen to be high enough for this species to be essentially homogeneous in the given domain size. As a default, the diffusivity of a local species is chosen to be zero, though in some cases we allow the species to be weakly diffusible (see Supplementary Information). A localised species is always non-diffusible. To understand the interplay of spatial organisation and the network, we assess lumped and spatially organised versions of the network, focussing particularly on input-ouput characteristics. This already reveals the basic impact of spatial organisation and is seen transparently for the nominal parameter values. In particular this interplay already reveals basic capabilities and constraints associated with spatial organisation. This is reinforced by a focussed analysis of spatial parameters, in particular, diffusivity of species and location (and size) of the localisation, to reveal further qualitative trends. Furthermore, multiple trends, associated with the variation of spatial parameters are observed across different motifs and different spatial realisations of a motif, and this further testifies to the robustness of our results. Finally in multiple instances, we reveal qualitative insights regarding new behaviour emerging, and this is consolidated by analytical work, explicitly revealing the interplay of factors giving rise to such behaviour. All in all, the analysis (in some cases computational, in other cases analytical, and in other cases constructive: see summary in Supplementary Table 1) transparently reveals the reasons for the observed behaviour and is indicative of the essential robustness of the conclusions.

The parameter values for all the results presented here are given in the Supplementary Information. The models for a given network are built up as described above, and the relevant code is presented in the Supplementary Information (note the notation used).

**Choice of candidate networks**. The description above allows for the construction of the model of a given spatial network (specific network, spatial characteristics of nodes, boundary conditions, parameters). We now comment on how we choose the networks which form the basis of our study. At the outset we emphasise that studying the interplay of spatial organisation and networks, entails the dissection of different networks on one hand, and different spatial realisations of the network on the other. This can be examined in two complementary ways: (i) imposing spatial organisation on a given network (Motif-centric approach) and (ii) imposing a pattern of interactions on components belonging to different spatial classes (space-centric approach). Taken together this complementary approach allows us to assess and analyze a variety of networks with different types of spatial organisation. It also creates a platform for further analysis in multiple contexts.

**Motif-centric approach**. In this approach we start with a set of two and three-node motifs involving a maximum of four interactions between them (maximum two interactions for the two-node motifs). This allows us to keep to a tractable level of complexity, while still allowing motifs that include multiple feedback loops, and combinations of feedforward and feedback loops. For this very reason, such motifs have been the focus on many studies in the temporal context[20]. The motifs in this chosen set are capable of exhibiting a wide range of behaviours—from simple signal transduction and adaptive behaviour, to combinations of multistability and oscillations in the purely temporal context, and self-organised spatio-temporal behaviour such as pattern formation and travelling waves.

Limiting ourselves to three nodes can however be quite restrictive in the context of imposing spatial organisation. For instance, localising two interacting nodes at different locations would prevent their direct interaction, thus breaking the motif. In order to overcome such constraints, and to allow us to examine cases with multiple localised components, we also introduce spatial organisation in the interactions themselves. In the basic case, a given interaction is allowed to occur at any location where the two interacting nodes are present. In addition, we will also examine the following possibilities: (i) the interaction is facilitated by a diffusing (global) species—this allows interactions between separated localised nodes to be maintained (see below), (ii) the interaction is itself localised, i.e. restricted to occur in one or more sub-domains; such a localised interaction between nodes can be facilitated in natural pathways, for example, by having an intermediate step (essentially an additional node) in the interaction, that involves a localised species, or if the interacting species are required to bind to a localised scaffold for the reaction to take place and (iii) the interaction involves a sequential combination of localised and diffusing components in the following way—if the network structure of the motif involves a node X regulating a node Y, then such an interaction is realised by introducing a pair of implicit nodes, say a localised node Z, and a global node W, and having X activate Z, which in turn activates W, which then regulates Y. Without node Z, this corresponds to (i) above, and without W, this corresponds to (ii) above.

All in all, imposing a spatial characteristic of the interaction can be regarded as spatially describing implicit nodes/intermediate steps in the interaction. This allows us to examine a broader class of spatial networks: (i) working within the initial three-node motif class (ii) allowing for more non-trivial spatial organisation, going beyond the basic restrictions imposed by the presence of only three nodes and (iii) avoiding the need to scan nodes of a higher number. This allows us to also reveal in a naturally constructive manner, how different types of spatial behaviour can emerge from small networks, many of which already have analogues in real biological systems.

In each case, we first characterise the behaviour of the motif in the purely temporal context (using the ODE model). Then we examine the different possible spatial realisations, assigning different nodes to different spatial classes (and also considering interactions in different spatial classes as outlined above, by incorporating implicit nodes). We consider a maximum of two different locations, where nodes may be localised. In a 1-D spatial domain, having two locations of localisation is sufficient, both to divide the whole domain into distinct regions, and to allow for basic effects such as source-sink separation, as well as delay effects arising from diffusion between locations.

**Space-centric approach**. A complementary approach to building and analyzing spatial networks is to start from the three different spatial classes (local, global and locallized) and building up a pattern of interactions (network) involving components from these three classes. In the case of multiple localisation, the distinct locations are treated as different spatial classes. In this manner we can visualise the system in terms of the spatial classes. Many networks built using this approach overlap with networks studied in the motif-centric approach. However, an advantage of the space-centric approach is that it allows us to naturally build up complex network patterns, framed by the spatial organisation at the outset. In particular, it also allows us to build networks which may not have been obtained easily from a network centric approach and additionally provides conceptual clarity. Most of the networks studied were easily constructed from both approaches.

**Comments on models.**

1. Our implementation of models has involved models developed in one spatial dimension. This allows us to focus on the most basic aspects of interplay between networks and spatial organisation. We point out that this, by its nature, is also relevant in higher dimensions. However there are additional aspects (e.g. effect of geometry) which necessitate a dedicated approach of its own, which is beyond the scope of the current study. The results presented here serve as a platform for such investigations. Some of the additional effects which emerge in higher dimensions are more complex patterns of localisation in the domain, more complex boundaries, a diversity of pattern formation, more scope for using localisation to control patterns. A recent study of multi-dimensional effects in symmetry-breaking and pattern formation in mass conserved systems is presented in ref. [40].

2. Our study focussed on a broad class of representative motifs, which have been used for studying information processing. Networks can exhibit different types of complexity with a multitude of nodes. Our analysis provides insights into a number of basic circuits whose relevance is demonstrated by the fact that (a) they are present in multiple cellular networks (exhibiting their characteristic behaviour), (b) they serve as network building blocks and (c) they also represent a broad range of basic qualitative behaviour encountered in concrete pathways and networks.

3. Other aspects of network behaviour, for instance the effect of intrinsic noise, and stochasticity is beyond the scope of the current study, but can draw from its insights. The study of noise in and its impact on information processing is an interesting theme (e.g. see ref. [41]). In our context, it is important to establish the implications of localisation in a deterministic setting first (and we have presented multiple insights in this regard). This would be a necessary platform for systematically exploring the role of stochasticity, which needs a detailed and dedicated study of its own.

4. A similar comment can be made about dynamic localisation. As discussed in the text, dynamic localisation is an important emerging theme (both in natural and engineered biology). Our results provide relevant insights into the eventual changes in behaviour following a change in localisation. However, a systematic exploration of dynamic localisation will involve detailed spatio-temporal studies which build on this study, and this will be done in the future.

5. Our models which involve basic depictions of localised, global and local species are relevant in cell-free settings, at the intracellular level and also to cells and populations. In the case of cells and populations, it depicts the interactions betweeen species which are globally diffusing (for e.g. in the extracellular environment), species localised to a specific set of cells in a given location/region (the cells themselves being immobile and stationary), and species present in all cells, but non-diffusing.

6. Our systems approach provides a way of exploring the role of localisation in different types of circuits. Given the current interest in bottom-up synthetic biology (including in cell-free systems) in engineering biochemical circuits with different functionalities[42–44], this approach can serve as a valuable addition to the existing toolkit.

7. The models in 1-D are relevant to describing dynamics on the cell membrane or along a cellular cross section.We point out that tools such as Virtual Cell can help simulate models with localisation in higher dimensions and with different regions of localisation therein. However starting out with computations in such a setting for the networks we examine, can make the computational analysis time-consuming and obscure certain key underlying patterns (our focal point). Having isolated key underlying cause-and-effect patterns, it becomes easier to incorporate factors such as the effect of dimensions or geometry.

**Spatial regulation of Polo Kinase: Models.** Here we present the basic aspects of the models used for the exemplar cases (also see Supplementary Note 7). We examine two distinct intracellular contexts where experimental evidence shows that the localisation of a protein, Polo kinase/Polo like kinase plays a central role. The first is in the transduction of cytoplasmic protein gradients that regulates polarisation in the *C. elegans* zygote. The second involves nuclear localisation of Polo kinase triggering a mitotic switch, through a mechanism involving multiple possible feedback interactions. Here we focus on a model suggested by experiments in *Drosophila*—however, we note that nuclear localisation of Polo and its role in the mitotic switch is observed across different organism[45]. Both these examples highlight multiple themes encountered in the paper: the effect of localisation, the presence of two diffusivity nodes, and consequences for inference as well as engineering design principles.

We first discuss the transduction of cytoplasmic protein gradients by PLK1 A cascade of intracellular protein gradients is established during asymmetric cell division in the *C. elegans* zygote[46]. The RNA binding protein Mex-5 forms a gradient, with higher levels in the anterior cytoplasm. This upstream gradient drives gradient formation in the Polo like kinase PLK-1. The PLK-1 gradient in turn drives gradient formation in the RNA binding protein Pos-1. Experimental evidence and modelling suggests that these gradients are formed by spatially graded interconversion between slow and fast diffusing forms of these proteins in each case[25].

We build a model of this cascade using nodes having interconverting fast and slow diffusing forms in a 1-D spatial domain with no-flux boundary conditions. This type of model has been used to describe an individual step of this cascade and fit to experimental data[25]. We parametrise our model to capture the qualitative features of these gradients shown experimentally. The structure of the model is as follows:

- Each step of the cascade—Mex5, PLK1 and Pos1 is represented by a node with interconverting fast and slow diffusing forms.
- At each node, one of the interconversion reactions (fast to slow diffusing forms OR slow to fast diffusing forms) is mediated by an upstream node (posterior localised PAR1 in the case of Mex5), while the reverse reaction is assumed to have a fixed rate constant that is uniform across the domain.
- Posterior localised PAR1 mediates Mex5 conversion from slow to fast diffusing form.
- The slow diffusing form of Mex5 mediates PLK1 conversion from fast to slow diffusing form.
- The slow diffusing form of PLK1 mediates conversion of Pos1 from slow to fast diffusing form.

The model equations are as follows, with $^*$ denoting the slow diffusing form:

$$\frac{\partial [Mex5^*]}{\partial t} = k_{01}[Mex5] - k_{02}[Mex5^*] - k_{PAR1}[Mex5^*] + D_{Mex5^*}\frac{\partial^2 [Mex5^*]}{\partial \theta^2}$$

$$\frac{\partial [Mex5]}{\partial t} = -k_{01}[Mex5] + k_{02}[Mex5^*] + k_{PAR1}[Mex5^*] + D_{Mex5}\frac{\partial^2 [Mex5]}{\partial \theta^2}$$

$$\frac{\partial [PLK1^*]}{\partial t} = k_{03}[PLK1] - k_{04}[PLK1^*] + k_3[Mex5^*][PLK1] + D_{PLK1^*}\frac{\partial^2 [PLK1^*]}{\partial \theta^2}$$

$$\frac{\partial [PLK1]}{\partial t} = -k_{03}[PLK1] + k_{04}[PLK1^*] - k_3[Mex5^*][PLK1] + D_{PLK1}\frac{\partial^2 [PLK1]}{\partial \theta^2} \quad (4)$$

$$\frac{\partial [Pos1^*]}{\partial t} = k_{05}[Pos1] - k_{06}[Pos1^*] - k_6[PLK1^*][Pos1^*] + D_{Pos1^*}\frac{\partial^2 [Pos1^*]}{\partial \theta^2}$$

$$\frac{\partial [Pos1]}{\partial t} = -k_{05}[Pos1] + k_{06}[Pos1^*] + k_6[PLK1^*][Pos1^*] + D_{Pos1}\frac{\partial^2 [Pos1]}{\partial \theta^2}$$

The spatial domain is divided into two sub-domains, representing the anterior and posterior cytoplasm. The rate constant $k_{PAR1}$ is zero outside of the posterior compartment.

Note that in the above model the slow diffusing forms of these proteins are the active forms with respect to downstream regulation.

We also examine an alternative realisation of the cascade, which reproduces the experimentally observed opposing relationship between the Mex5 and Pos1 gradients, while reversing the PLK1 gradient. In this case:

- The slow form of Mex5 mediates PLK1 conversion from slow to fast diffusing form.
- The slow form of PLK1 mediates conversion of Pos1 from fast to slow diffusing form.

Examining both realisations with a consistent relation between the input and output nodes is done for multiple reasons: (1) it allows us to elucidate basic design principles associated with gradient transduction across multistep cascades, (2) since Polo kinase is a focal point, it also examines different ways in which Polo kinase may be regulated and how it affects the overall outcome.

We now turn to a mathematical investigation of a proposed network governing mitotic entry in *Drosophila* with a focus on the role of Polo therein. The mathematical model is based on a number of experiments which have been consolidated into a proposed network discussed in[47]. The goal of this work is to analyse and evaluate the proposed network model and reveal the underlying design principles. The model involves inactive Polo initially localised in the cytoplasm (sequestered by microtubule bound Map205), and being transferred to the nucleus upon activation. In the nucleus, Polo activates Cdc25 (initially localised in the nucleus). Cdc25 activation triggers its transport to the cytoplasm, where it activates the CycB-Cdk1 complex. This triggers localisation of CycB-Cdk1 in the nucleus, where it can further enhance activation of Cdc25 by Polo. Thus, Cdc25 and CycB-Cdk1 form a spatially distributed positive feedback loop, which potentially allows a sharp, irreversible mitotic transition.

Interconversion between forms with different mobility has been used to study nuclear localisation of CycB-Cdk1 in the context of a cell-cycle switch[4]. We adopt a similar approach here, using nodes with interconverting fast and slow diffusing forms to represent the different spatially regulated components of this network, in a 1-D spatial domain with no-flux boundary conditions. Note that we do not explicitly describe the interface between nuclear and cytoplasmic compartments, as the essential insights emerge are the same, and the functioning of the spatially distributed network does not depend on this in any essential way.

The structure of the model is as follows:

- The 1-D spatial domain is divided into two sub-domains, representing the cytoplasm and the nucleus
- Polo is converted from fast (active) to slow diffusing (inactive) form in the cytoplasm
- The slow diffusing (inactive) form of Polo can be sequestered by binding to Map205, which is localised in the cytoplasm
- The slow diffusing (inactive) form of Polo can be converted to the fast diffusing (active) form by an upstream signal, representing Aurora B

- Cdc25 is converted from fast (active) to slow (inactive) diffusing form in the nucleus
- Active Polo in the nucleus converts Cdc25 from slow to fast diffusing form
- CycB-Cdk1 can exist in three forms, a fast diffusing inactive form, a fast diffusing active form, and a slow diffusing active form
- Active, fast diffusing CycB-Cdk1 can be converted to active slow diffusing form in the nucleus. This conversion involves a positive feedback, as suggested by[4].
- Active Cdc25 in the cytoplasm can convert inactive CycB-Cdk1 to the fast diffusing active form
- Active CycB-Cdk1 in the nucleus promotes Cdc25 activation by Polo

We use the following notation in the model equations:

- $Polo^*$ denotes the slow diffusing, inactive form of Polo
- $Polo^{bound}$ denotes the inactive form of Polo bound to Map205 (non-diffusible)
- $Polo$ denotes the fast diffusing active form of Polo
- $Cdc25^*$ denotes the slow diffusing inactive form of Cdc25
- $Cdc25$ denotes the fast diffusing active form of Cdc25
- $CycB - Cdk1^{**}$ denotes the slow diffusing active form of CycB-Cdk1
- $CycB - Cdk1^*$ denotes the fast diffusing active form of CycB-Cdk1
- $CycB - Cdk1$ denotes the fast diffusing inactive form of CycB-Cdk1

The model equations are as follows:

$$\frac{\partial [Polo^{bound}]}{\partial t} = -k_{unbind}[Polo^{bound}] + k_{bind}[Polo^*]([Map205_{Total}] - [Polo^{bound}]) + D_{Polo^{bound}}\frac{\partial^2 [Polo^{bound}]}{\partial \theta^2}$$

$$\frac{\partial [Polo^*]}{\partial t} = k_{unbind}[Polo^{bound}] - k_{bind}[Polo^*]([Map205_{Total}] - [Polo^{bound}])$$
$$+ k_{-Polo}[Polo] - k_{02}[Polo^*] - k_{AuroraB}[Polo^*] + D_{Polo^*}\frac{\partial^2 [Polo^*]}{\partial \theta^2}$$

$$\frac{\partial [Polo]}{\partial t} = -k_{-Polo}[Polo] + k_{02}[Polo^*] + k_{AuroraB}[Polo^*] + D_{Polo}\frac{\partial^2 [Polo]}{\partial \theta^2}$$

$$\frac{\partial [Cdc25^*]}{\partial t} = k_{03}[Cdc25] - k_{04}[Cdc25^*] - k_3[Polo][Cdc25^*] + D_{Cdc25^*}\frac{\partial^2 [Cdc25^*]}{\partial \theta^2}$$

$$\frac{\partial [Cdc25]}{\partial t} = -k_{03}[Cdc25] + k_{04}[Cdc25^*] + k_3[Polo][Cdc25^*] + D_{Cdc25}\frac{\partial^2 [Cdc25]}{\partial \theta^2}$$

$$\frac{\partial [CycB - Cdk1^*]}{\partial t} = k_{05}[CycB - Cdk1] - k_{06}[CycB - Cdk1^*] + k_5[Cdc25][CycB - Cdk1]$$
$$+ k_{07}[CycB - Cdk1^{**}] - k_8[CycB - Cdk1^*] + D_{CycB-Cdk1^*}\frac{\partial^2 [CycB - Cdk1^*]}{\partial \theta^2}$$

$$\frac{\partial [CycB - Cdk1^{**}]}{\partial t} = -k_{07}[CycB - Cdk1^{**}] + k_8[CycB - Cdk1^*] + D_{CycB-Cdk1^{**}}\frac{\partial^2 [CycB - Cdk1^{**}]}{\partial \theta^2}$$

$$\frac{\partial [CycB - Cdk1]}{\partial t} = -k_{05}[CycB - Cdk1] + k_{06}[CycB - Cdk1^*] - k_5[Cdc25][CycB - Cdk1]$$
$$+ D_{CycB-Cdk1}\frac{\partial^2 [CycB - Cdk1]}{\partial \theta^2}$$

$$(5)$$

The spatial domain is divided into two sub-domains, representing the nucleus and cytoplasm. $D_{Polo^{bound}}$ is set to zero. Spatial constraints are imposed on the following parameters and kinetic rate constants:

- $k_{bind}$ is zero in the nucleus.
- $[Map205_{Total}]$ is zero in the nucleus.
- $k_{-Polo}$ is zero in the nucleus.
- $k_{03}$ is zero in the cytoplasm.
- $k_3$ is a function of the total active CycB-Cdk1 in the nucleus: $k_3 = k_a + k_b([CycB - Cdk1^*] + [CycB - Cdk1^{**}])$. $k_3$ is zero in the cytoplasm.
- $k_8$ is a function of the total active CycB-Cdk1 in the nucleus: $k_8 = \frac{k_p([CycB - Cdk1^*] + [CycB - Cdk1^{**}])^4}{K_p + ([CycB - Cdk1^*] + [CycB - Cdk1^{**}])^4}$, $k_8$ is zero in the cytoplasm.

We also examine a possible negative feedback regulation of Polo by Optn similar to that proposed by[48], as follows (see Supplementary Information):

- Optn is converted from fast (active) to slow (inactive) form in the cytoplasm.
- Active Polo in the cytoplasm converts Optn slow (inactive) to fast (active) form in the cytoplasm.
- Active Optn can convert fast (active) Polo to slow (inactive) form in the nucleus.

We comment on the choice of parameters in the models for the two exemplar cases involving Polo Kinase above (also see Supplementary Information). Our goal in each case is to address questions of a qualitative nature. In the case of gradient transduction we focus on the qualitative nature of gradient transduction, and in particular reveal sources of qualitative difference in two architectures of the cascade: this does not depend on any particular parameter set, but emerges from a basic analysis of the building blocks of the cascade and their interaction. In the cell-cycle example, we use the models to evaluate certain key postulations about network interactions and their qualitative impact (the focal point in current biological studies): the choice of our parameters is sufficient for this, and our insights again transparently emerge from the network structure

**Reporting summary**. Further information on research design is available in the Nature Research Reporting Summary linked to this article.

## Data availability
The authors declare that the data supporting the findings of this study are all contained in the main text and Supplementary Information. Data supporting the findings of this paper are available from the corresponding author upon reasonable request.

## Code availability
We have made the codes for the study available in the following way. The models for all our studies are available in the Main Text or Supplementary Information. We have included MATLAB segments for the variations of models which we use in the Supplementary Information. The full code (i.e. full set of equations in MATLAB format) is presented for a subset of cases (this is because many cases involved changes within a basic code). We have included one full code for each of the subsections 2.1–2.4 in the Supplementary Information. The MATLAB files containing this code are also uploaded on a GitHub repository at the following link: https://doi.org/10.5281/zenodo.4944672. From the existing full code, and the different models presented (in MATLAB format), codes for the various cases can be recreated. Furthermore, anyone interested in further details can contact the corresponding author.

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

## Acknowledgements

We gratefully acknowledge funding to G.M. through a Departmental Scholarship at the Department of Chemical Engineering, Imperial College London.

## Author contributions

J.K. formulated the problem and the approach and framework for analysis, J.K. and G.M. developed this in multiple directions, G.M. performed the computational analysis of the models and the associated code development, G.M. performed analytical work to analyze the models, with input from J.K., G.M. and J.K. analyzed the data and interpreted the results, J.K. wrote the main text with input from G.M., G.M. wrote the Supplementary Information with input from J.K.

## Competing interests

The authors declare no competing interests.
