## [Peer Review File · Nature Communications]

REVIEWER COMMENTS

Reviewer #1 (Remarks to the Author):

Menon and Krishnan presented a very thorough analysis of the effects of localization on signal processing properties of regulatory network motifs. They show how the combined effects of diffusion and boundary conditions can dramatically change the operating regimes. The work is well motivated. My only comment is that all results from 1d PDE systems. The authors should at least mention higher dimensional effects, e.g. those that lead to intracellular symmetry breaking in mass-conserved problems.

<https://pubmed.ncbi.nlm.nih.gov/29933887/>

Reviewer #2 (Remarks to the Author):

In the manuscript titled "Spatial localization meets biomolecular networks", the authors elucidated the interplay of networks and spatial organization, which is an interesting topic. Focusing on the effect of spatial localization on the components of a series of theoretical network models, the authors showed some of the impacts of spatial localization on network function and behavior. However, it is difficult to get the main point of this work, and too many issues or results delute the main conclusion.

1. Diffusivities certainly affected many aspects of dynamical systems, which is not new. Actually, many works adopted the time delay to approximate their effects, and similar conclusions were drawn.
2. What is the systems framework (or algorithm or model, not abstract concept) developed by this manuscript, claimed by authors ? It is unclear what is the specific framework, which can be applied to analyze the biological (omics or other) data. authors should provide the detail procedures.
3. The exhibited cases were under relatively ideal situations and the networks were simple. Authors should show a case based on the real biological data, to solve a specific biological problem or correct the wrong results.
4. Are those circuits illustrated under respective conditions are robust? How is the effect of strong noise on the results ?
- 5) A number of parameters are used in the algorithm. Although the parameters of the model have been discussed in P23-P24, it is still unclear how to determine and choose the specific values for these parameters?

Minor:

(1) The code should be provided so that one can check the applicability.

(2) There are typos that should be revised, e.g., line 11 in page 22,

(3) There are too many grammar mistakes. In page 27, "Our models which involve basic depictions of localized, global and local species is relevant in cell-free settings".

In page 27, "they are present in a number of real networks", and so on.

(4) Figures should be polished. In Fig.4E, "Inhibition" was partly covered.

(5) Some relevant references are missing, e.g., Qiao et al., 2019, Cell Systems 9, 1–15, 2019. <https://doi.org/10.1016/j.cels.2019.08.006>. Also two references are incomplete: 1. Reference "Hegde, R. S. & Zavodszky, E. 2019." lacks page information. 2. Reference "Lee, M. J et al., 2019." lacks issue information.

(6) Some sentences are ambiguous. E.g. "Our analysis provides insights into a number of basic circuits whose relevance is demonstrated by the fact that (a) they are present in a number of real networks (exhibiting their characteristic behaviour), (b) serve as network building blocks, and (c) also represent a broad range of basic qualitative behaviour encountered in concrete pathways and networks." No subjects for (b) and (c).

(7) Why do the authors only use constant localization intensity? The intensity of localization is considered to change dynamically in the biological system.

(8) Many spelling errors:

1. Page 2, paragraph 4, line 2 : "enties" should be "entities".
2. Page 3, paragraph 1, line 10 : "intracelllar" should be "intracellular".
3. Page 4, paragraph 4, line 5 : "specfication" should be "specification".
4. Page 5, paragraph 3, line 6 : "noreworthy" should be "noteworthy".
5. Page 8, paragraph 3, line 11 : "by built" should be "be built".
6. Page 10, paragraph 1, line 9 : "facillitate" should be "facilitate".
7. Page 10, paragraph 3, line 1 : "spaial" should be "spatial".
8. Page 14, paragraph 2, line 5 : "implemented" should be "be implemented".
9. Page 21, paragraph 2, line 1 : "coefficients" should be "coefficients".
10. Page 24, paragraph 1, line 15 : "constaints" should be "constraints".

Response to Reviewer 1

We thank the reviewer for a careful reading of the manuscript and for the comments. We are pleased to see the reviewer find the work thorough and also well-motivated. Our intention was to reveal multiple facets of the interplay of localization and networks, and that informed the way we approached the paper: we are therefore happy to see that this finds consonance with the reviewer.

My only comment is that all results from 1d PDE systems. The authors should at least mention higher dimensional effects, e.g. those that lead to intracellular symmetry breaking in mass-conserved problems.

<https://pubmed.ncbi.nlm.nih.gov/29933887/>

Regarding the comment about the effect of multiple dimensions and the model being 1-D PDEs—we had discussed this briefly in the Models and Methods Section (page 24). We expand on that to discuss the relevance of our results there and the additional things which could be expected in multiple dimensions (page 26: Comments on models, point 1). The new reference on mass conserved systems and intracellular symmetry breaking is also incorporated.

We also summarize the main additions/changes to the manuscript

The main changes are

- (i) The addition of a short section (Section 2.5) along with a new figure (current Figure 9) to address Reviewer 2's comment about analysing a concrete biological system. We focus on two pathways/processes involving the regulation of Polo-kinase, where localization plays a critical role—a cascade responsible for polarization in *C. elegans* and a circuit responsible for a switch leading to mitotic exit in *Drosophila*. Both systems can be understood in terms of circuits which contain two-diffusivity nodes. In each case there are a number of experiments performed, and postulated hypotheses/networks underlying their function. We develop mathematical models to evaluate the postulated hypotheses/networks, address existing questions, along with other questions of our own. In each case we analyse the system, by starting with the basic building blocks, and using this as a basis for evaluating the circuit behaviour and the role of localization therein. We also reveal underlying design principles/constraints regarding the functioning of these circuits.
- (i) An associated augmentation to the Models and Methods.
- (ii) Trimming of a few sections to reduce the number of words: in particular (a) Reducing the survey of previous work to a much more concise discussion (with additional details and references in supplementary). This we believe, strikes a balance between providing enough of a backdrop and justification of the work (something appreciated by Reviewer 1) and not expending too many words on what is ultimately previous work (b) Presenting in a slightly more concise manner,

- the leadup to the results in Section 2, eliminating repetition and any unnecessary points
- (c) Trimming the initial part of the conclusions, for the same reasons
- (iii) Section 2.3 illustrated a case of how localization could lead a negative feedback circuit to be inferred as a positive feedback circuit. We have slightly expanded and consolidated this analysis to illustrate the underlying principle/concept, and discuss under which conditions such an incorrect inference can be made. This has led to 3 additional panels in Fig. 7 and an extra paragraph discussing these points. This also has implications for the specific biological context we study and we explicitly refer to it there.
- (iv) We have made the codes for the study available. We have included MATLAB segments for all variations of models which we use. The full code is presented for a subset of cases (this is because many cases involved changes within a basic code). From the existing full code, and the different models presented (in MATLAB format), codes for the various cases can be recreated. Furthermore, anyone interested in further details can contact me (Krishnan) as corresponding author.

Response to Reviewer 2:

We thank the reviewer for a careful reading of the manuscript and the comments. A summary of the main changes in the text is provided at the end of the response to the reviewer's comments.

In the manuscript titled “Spatial localization meets biomolecular networks”, the authors elucidated the interplay of networks and spatial organization, which is an interesting topic. Focusing on the effect of spatial localization on the components of a series of theoretical network models, the authors showed some of the impacts of spatial localization on network function and behavior. However, it is difficult to get the main point of this work, and too many issues or results delute the main conclusion.

We are pleased to see that the reviewer comments on the interplay of networks and spatial organization being an interesting topic. We address the main point (last sentence of the comment)

Focal point of the work: We first note that we focus on a specific, ubiquitous aspect of spatial organization, namely localization (many associated references have been presented). The main point is that (i) to understand the impact of localization, one needs to examine its presence in networks where nodes may have different spatial characteristics (ii) uncovering the diversity of ways in which localization can impact networks, necessitates systems approaches which transcend a particular context (iii) Our focus is explicitly to reveal multiple facets and dimensions to the interplay of localization and networks. This is why we address alterations in network behaviour, impact on pattern formation, consequences for inference and potential for engineering. We believe that focussing on the diversity at the outset allows us to unravel what localization may contribute in many contexts, the different guises in which it may affect behaviour and how it could be actively exploited. This allows to connect with multiple strands of work in systems and synthetic biology.

Multifaceted nature of study: The value of a multifaceted study can be seen when we consider concrete cases (in response to the Reviewer comment—see below). This connects to Section 2.1 (Effect of localization on networks), Section 2.3 (Localization and Inference) and Section 2.4 (Localization and Engineering: especially in the context of negative feedback having anomalous behaviour)

The multiplicity of results (still organized along four focussed themes) is a consequence of this. Thus the reader should not expect one key cause-effect underlying all the results because there is a fundamentally diverse way in which localization impacts networks and we aim to unravel key aspects of this.

Transparency of cause and effect: Furthermore in each case we aim to provide a transparent cause and effect, whether through analytical or computational work and motifs which are either basic building blocks, motifs studied elsewhere or circuits which are constructed to illustrate specific points (a sentence emphasizing this is added in the Models Section: end of page 23). This aspect is discussed further (at the end of response to main comments, in “Nature of Analysis and the Results”).

We also incorporate some changes in the manuscript to address this comment, and make it easier for the reader to follow this line of thinking. In Fig. 10 (the old Fig. 9), we now provide an expanded caption where we provide a big picture view of the entire paper to show how everything holds together.

1. Diffusivities certainly affected many aspects of dynamical systems, which is not new. Actually, many works adopted the time delay to approximate their effects, and similar conclusions were drawn.

We certainly realize that diffusivity can affect multiple aspects of dynamical systems. However, our focus is not on the effect of diffusivity per se, it is really on the impact of localization and in that context, the role of diffusivity of other species needs to be assessed. Thus we not only discuss and highlight effects on diffusivity in specific places, but actually address the impact of localization and the diffusivity of other species (high or low). We have examined different possibilities of diffusivities of various nodes in every network (whether local or global) and this is itself a much more focussed network oriented study of the role of different diffusivities in networks with localization. Finally, we examine networks with two-diffusivity nodes—the investigation of such mass-conserved systems is of considerable current interest (eg. Halatek and Frey, *Nature Physics*, 2018, Diegmiller et al, *Biophysical Journal*, 2018, Wu et al, *PNAS*, 2018).

We have made a rephrasing to emphasize the fact that when we discuss diffusivities in the Text (Section 2.1), we are studying this in the context of localization. We can summarize this by saying we bring in localization (the central focus throughout the paper) and assess different diffusion characteristics of the various nodes as part of our study.

We reiterate the point that the paper is about the diverse consequences of localization (enabling new behaviour, trapping of fronts, enabling pattern formation, key regulators of patterns without distorting them, complicating the inference process, impacting homeostasis and as a tool for rewiring and design of networks).

Diffusivity effects and delays: Regarding the point that many papers use a delay to approximate the effect of diffusivities, we emphatically state that the many insights we obtain (see previous paragraph) go well beyond a simple delay effect. To start with, many results focus on steady states. Secondly going through case by case, the various studies of localization, there are in fact very few where the primary effect of diffusivity/localization is through the contribution of delays (and even if they are, this may not be represented as a delay term in an ODE). Things get even more complicated when one looks at a diverse range of networks (even network motifs) and this strongly confirms that delay in most cases is not the issue (even if a delay was incorporated, it would not be able to correctly account for many of the new phenomena uncovered)

In terms of the results, the result which may be most closely related to delays is the obtaining of oscillations with localization in two locations (which is in fact mostly discussed in Supplementary Material). The destabilization of uniform steady states to give inhomogeneous steady states is not appropriately described by a time-delay. Most other results are steady state results, and introducing time -delays into an ODE will not produce such results. Finally,

introducing a time-delay into an ODE (to serve as a proxy for the effects of transport, for instance) will result in an explicit description of space being ignored, which is a basic point of interest for many aspects of the paper.

2. What is the systems framework (or algorithm or model, not abstract concept) developed by this manuscript, claimed by authors ? It is unclear what is the specific framework, which can be applied to analyze the biological (omics or other) data. authors should provide the detail procedures.

We use the word framework in the following sense: (Cambridge Dictionary): “a system of rules, ideas or beliefs to plan or decide something” or “a basic structure underlying a system, concept or text” Our goal is to evaluate the impact of localization and to that end we create a framework where we consider (a) candidate networks and (b) different ways in which spatial localization can be incorporated amidst other species which are either highly diffusible or weakly diffusible. We also use two different ways of building up such networks. This collection of networks and imposition of localization and ways of constructing and analysing such spatial networks is what our framework involves.

Our framework is not an algorithm for connecting with omics data (nowhere do we claim that!), and is not a single model (though everything could be considered a subset of some larger model—that is neither useful nor necessary for what we do). Our framework is aimed at connecting localization and networks, and in so doing we evaluate many networks with localization.

Our approach for building up these networks is also useful in specific contexts where one or more candidate networks and/or localization patterns may be desired to be evaluated. That can build on the approach we have used here, but will necessarily have to engage with and incorporate particular details of the relevant context (rather than use something directly “off the shelf”).

Our framework (in the sense of the word we have used), has helped us systematically examine different aspects of the concrete cases we have studied in a structured way (in Section 2.5).

In our trimming of the conclusions, we had removed certain sentences, and also edited others. One sentence mentions the word “framework” in a certain context, which is perhaps what might have caused some confusion. We have checked the use of the word “framework” through the paper to ensure that it is consistent with the way we intended it, with no scope for confusion.

3. The exhibited cases were under relatively ideal situations and the networks were simple. Authors should show a case based on the real biological data, to solve a specific biological problem or correct the wrong results.

We have added a short section (Section 2.5) along with a new figure (current Figure 9) to address this comment about analysing a concrete biological system.

We focus on two pathways/processes involving the regulation of Polo-kinase, where localization plays a critical role—a cascade responsible for polarization in *C. elegans* and a

circuit responsible for a switch leading to mitotic exit in *Drosophila*. There are a range of relatively recent experiments and data in both these contexts, along with postulated circuits governing the observed behaviour. In the case of *C. elegans*, experiments and modelling have been used to study some basic aspects of one step of the cascade, with open questions regarding the behaviour of the overall cascade and the role of localization therein. In the case of *Drosophila*, there is a postulated circuit with both positive and negative feedback based on existing experimental data, with a range of questions emerging of when and how the circuit would provide the observed outcome and what the role of the negative feedback is.

Both systems can be understood in terms of circuits which contain two-diffusivity nodes. In each case there are a number of experiments performed, and postulated hypotheses/networks underlying their function. We develop mathematical models to evaluate the postulated hypotheses/networks, address existing questions, along with other questions of our own, which are focussed on localization. In each case we analyse the system, by starting with the basic building blocks, and using this as a basis for evaluating the circuit behaviour and the role of localization therein. We also reveal underlying design principles/constraints regarding the functioning of these circuits

These studies contribute to the elucidation of these systems, answering specific questions, and provide non-trivial insights from a distinct perspective.

We mention, in passing, that we are also involved in collaborative work in other specific cellular biological contexts (tumours) where localization plays a key role (but that investigation is the topic of another paper being prepared) and so are keenly aware of the impact of localization in specific contexts as well.

4. Are those circuits illustrated under respective conditions are robust? How is the effect of strong noise on the results ?

Regarding the circuits used and their robustness. The underlying network circuits are chosen because they represent different widely observed behaviour. The reasons for the observed behaviour are well understood and well-studied. Thus we start with circuits which represent well studied behaviour for reasons which are well understood. We have chosen representative parameters in these circuits which are in the “middle” of the parameter region where the behaviour is observed. In some cases we also have analytical work which provides additional characterization of these circuits.

We then explore the effect of localization by considering localizing some nodes and having some nodes global. We use representative parameters for the size of the localized domain.

The nature of the results and how they are established: The essential point however is the nature of results we draw and how we establish it. Right through the paper, our approach is to establish transparent insights where the cause-effect relationship is clear. To do this, we use both analytical and computational work, and examine multiple circuits: basic motifs, motifs studied in the literature and some constructed circuits (to illustrate some key capabilities)

. In some cases the very existence of new behaviour is itself noteworthy, and we report it as such. Having done a lot of parametric analysis, we find that there are multiple parameter sets which readily represent the behaviour. In other cases, such as destabilization leading to pattern formation we perform analysis to show exactly why this is possible. This is also

relevant to our insights about hybrid pattern forming systems, incorrect inferences in feedback, breakdown of homeostasis, engineering circuits which buffer against spatial separation. In each of these instances, supporting analytical work along with basic computation transparently reveals the factors at play. In other cases, the analysis is computational, but again the insights emerge for a very transparent reason. At the end of the response to main comments (“Nature of analysis and the results”), we provide a list of results we have, supporting the point that the insights are transparent (and hence the conclusions are clear-cut). This is also included as an additional table in Supplementary Material.

All this testifies to the essential robustness of the conclusions and why they may be encountered in multiple instances.

Regarding the role of strong noise: the stochastic dimension to such studies is important. Here we note that (i) combining spatial and stochastic aspects leads to a considerably greater complexity (ii) the tools available for such analysis are fewer (iii) the effect of stochasticity necessarily needs the deterministic counterpart to be well understood first to serve as an orientation for exploration. This has already been encountered in temporal (lumped) networks, where stochastic studies could be fruitfully undertaken once the deterministic studies were thoroughly performed (iv) consequently it is important to perform such an investigation in the order of spatial (deterministic) followed by stochastic rather than the other way around (because space and localization impacts behaviour in many diverse ways, as we have seen) (v) such a study is well beyond the scope of the current study and could be the focus of one or more publications of their own.

We have added further comments in the Models and Methods (“Comments on models”, page 26, point 3) in relation to this.

5) A number of parameters are used in the algorithm. Although the parameters of the model have been discussed in P23-P24, it is still unclear how to determine and choose the specific values for these parameters?

We summarize the way we choose the parameters. For the basic network, the parameters are simply chosen to represent the basal behaviour of the circuit (eg adaptation, bistability, oscillations etc). In all these cases, the circuit has been studied and is well understood as is the reason for the behaviour observed. In some cases analytical work makes some of this even more explicit. We simply choose representative parameters in the “middle” of the parameter region where such behaviour is exhibited, so that the system is not near any bifurcation point. In essence, all we require from the basal network is that it demonstrates the behaviour it is supposed to (for reasons well understood) and that it has the structure that it has. A sentence emphasizing this has been added to the parameters section (end of first paragraph “Parameters” page 23)

In other cases, our results point to the creation of fundamentally new behaviour. Here analytical work establishes that the behaviour isn’t present in the network to start with. With regard to spatial parameters, the other parameters are the size of the localized domain. We choose a representative value, but all the results we obtained are observed even when this changes—and in many cases analytical work, and in others computational work transparently reveals the reason why the behaviour is obtained (see the comments below on “Nature of analysis and the results”)

We focus on local and global nodes generally—in some cases we vary the diffusivity to make some specific points related to diffusivity related behaviour and in all those cases we note it explicitly as such

Nature of analysis and the results:

While we examine multiple motifs and circuits with localization and draw out a diverse set of results, they do emerge for clear cut reasons. Here is a representative list

Section 2.1

(a) Basic effects of localization:

Analysis based on simple cause and effect and basic computation to reveal basic conclusions

(b) Localization and the role of diffusivity for bistable and oscillatory circuits, as well as multiple region localization for incoherent feedforward circuits:

Analysis: Basic computation supported by analytical work in multiple instances, explicitly revealing the factors at play

© Localization enables new behaviour:

Analysis: While the new behaviour is shown computationally (and can be seen in different parameter sets), the main point is that is something you do not see in the lumped system (ODE)—and this is established analytically

(d) Localization allows for translating network behaviour into spatial outcomes

Analysis: This is based on computational analysis of a circuit constructed and relies on the trapping of fronts, a phenomenon which is established. The novelty here is the information processing angle and what localization can do to enable this behaviour

Section 2.2

(a) Localization and generation of patterns:

Analysis: Computational results show that pattern formation can be readily obtained. Supporting analytical work through compartmental models reveals the underlying factors at work, explicitly

(b) Position of localization and symmetry breaking

Analysis: Computational work supported by basic analytical work to reveal the underlying functioning and reasons for behaviour

© Control and Selection of Patterns

Analysis: The analysis is computational, but the circuit is constructed based on basic intuition regarding enabling or preventing certain patterns for a very basic reason which is discussed in Supplementary Material

Section 2.3

(a) Basic effects

Analysis: Computations on basic models, complemented by analytical work

(b) Incorrect inference of feedback circuit

Analysis: Computation and analytical work showing that with localization the behaviour of the system is opposite to that of the ODE model (describing the nominal feedback structure). This has been further generalized with additional computation and analytical work

Section 2.4

(a) Impact of localization on homeostatic circuits

Analysis: Transparent revealing of cause and effect via computations and supporting analytical work

(b) Buffering against distance

Analysis: A circuit constructed based on analytical work, and confirmed computationally

© Rewiring enabling modular combination

Analysis: A basic set of circuits was chosen and computation shows that modular combination can be achieved. This is ultimately based on the idea that different circuits can be combined “downstream”, bypassing the kinetic constraints which prevent both behaviour being seen (Constructive approach).

(d) Alleviation of kinetic constraints by local elevation

Analysis: Computation and analytical work on basic prototype circuits: ultimately this relies on a basic feature of elevation of concentration—when that can and cannot help alleviate the kinetic constraint. The elevation of concentration of specific species (nodes) enables thresholds to be crossed.

Minor:

(1) The code should be provided so that one can check the applicability.

The equations and codes (to be used with a standard ODE solver in MATLAB) are now proved as follows (i) The equations for all models considered are available as equations in MATLAB syntax. This contains diffusion terms which are computed in a separate subroutine and that is also provided (ii) The various cases have been investigated by making changes to a small number of core files (eg adding or removing interactions). Keeping this in mind, we have provided the full files for a few representative cases (4—one for each subsection 2.1-2.4), and using this and the full equations supplied, the codes for every case can be generated. (iii) In case of further questions, the corresponding author is happy to provide more input and details

(2) There are typos that should be revised, e.g., line 11 in page 22,

This is now corrected

(3) There are too many grammar mistakes. In page 27, “Our models which involve basic depictions of localized, global and local species is relevant in cell-free settings”. In page 27, “they are present in a number of real networks”, and so on.

We have looked through the manuscript to eliminate any grammatical errors. We were not sure what was being referred to in the second example mentioned above, but have rephrased the sentence

(4) Figures should be polished. In Fig.4E, “Inhibition” was partly covered.

This was corrected. Multiple figures were polished, and an extensive caption provided for Fig. 9

(5) Some relevant references are missing, e.g., Qiao et al., 2019, Cell Systems 9, 1–15, 2019. <https://doi.org/10.1016/j.cels.2019.08.006>. Also two references are incomplete: 1. Reference “Hegde, R. S. & Zavodszky, E. 2019.” lacks page information. 2. Reference “Lee, M. J et al., 2019.” lacks issue information.

This reference is now included and referenced in the models and methods. The other two references are corrected

(6) Some sentences are ambiguous. E.g. “Our analysis provides insights into a number of basic circuits whose relevance is demonstrated by the fact that (a) they are present in a number of real networks (exhibiting their characteristic behaviour), (b) serve as network building blocks, and (c) also represent a broad range of basic qualitative behaviour encountered in concrete pathways and networks.” No subjects for (b) and (c).

This is corrected

(7) Why do the authors only use constant localization intensity? The intensity of localization is considered to change dynamically in the biological system.

We do recognize the importance of dynamic localization, and that is also the reason we ourselves have made reference to this.

In terms of studying dynamic localization: (i) A study of fixed (static) localization itself present a number of consequences which we draw out through the paper (ii) these insights provide many relevant conclusions for the steady state behaviour in systems where there is dynamic localization (eg. a transition from one localization pattern to another, or the sudden creation of a localization pattern) © In other cases, where the eventual state attained depends on the history of the dynamic localization or when the full spatiotemporal dynamics is desired, that needs a thorough systematic study of its own, using the above results as a platform. This is analogous to studies of transient dynamics in networks which are fruitfully pursued (thoroughly), once the steady state behaviour is properly characterized. This in turn is due to the fact that the transient dynamics relies on the interplay of both the existing systems and the temporal driving signal, and everything that entails.

(8) Many spelling errors:

1. Page 2, paragraph 4, line 2 : “enties” should be “entities”.
2. Page 3, paragraph 1, line 10 : “intracelllar” should be “intracellular”.
3. Page 4, paragraph 4, line 5 : “specfication” should be “specification”.
4. Page 5, paragraph 3, line 6 : “noreworthy” should be “noteworthy”.
5. Page 8, paragraph 3, line 11 : “by built” should be “be built”.
6. Page 10, paragraph 1, line 9 : “facillitate” should be “facilitate”.
7. Page 10, paragraph 3, line 1 : “spaial” should be “spatial”.
8. Page 14, paragraph 2, line 5 : “implemented” should be “be implemented”.
9. Page 21, paragraph 2, line 1 : “coefficients” should be “coefficients”.
10. Page 24, paragraph 1, line 15 : “constaints” should be “constraints”.

We apologize for this and have corrected it.

We also summarize the main additions to the text

The main changes are

- (i) The addition of a short section (Section 2.5) along with a new figure (current Figure 9) to address Reviewer 2's comment about analysing a concrete biological system. We focus on two pathways/processes involving the regulation of Polo-kinase, where localization plays a critical role—a cascade responsible for polarization in *C. elegans* and a circuit responsible for a switch leading to mitotic exit in *Drosophila*. Both systems can be understood in terms of circuits which contain two-diffusivity nodes. In each case there are a number of experiments performed, and postulated hypotheses/networks underlying their function. We develop mathematical models to evaluate the postulated hypotheses/networks, address existing questions, along with other questions of our own. In each case we analyse the system, by starting with the basic building blocks, and using this as a basis for evaluating the circuit behaviour and the role of localization therein. We also reveal underlying design principles/constraints regarding the functioning of these circuits.
- (ii) An associated augmentation to the Models and Methods.
- (iii) Trimming of a few sections to reduce the number of words: in particular (a) Reducing the survey of previous work to a much more concise discussion (with additional details and references in supplementary). This we believe, strikes a balance between providing enough of a backdrop and justification of the work (something appreciated by Reviewer 1) and not expending too many words on what is ultimately previous work (b) Presenting in a slightly more concise manner, the leadup to the results in Section 2, eliminating repetition and any unnecessary points (c) Trimming the initial part of the conclusions, for the same reasons
- (iv) Section 2.3 illustrated a case of how localization could lead a negative feedback circuit to be inferred as a positive feedback circuit. We have slightly expanded and consolidated this analysis to illustrate the underlying principle/concept, and discuss under which conditions such an incorrect inference can be made. This has led to 3 additional panels in Fig. 7 and an extra paragraph discussing these points. This also has implications for the specific biological context we study.
- (v) We have made the codes for the study available. We have included MATLAB segments for all variations of models which we use. The full code is presented for a subset of cases (this is because many cases involved changes within a basic code). From the existing full code, and the different models presented (in MATLAB format), codes for the various cases can be recreated. Furthermore, anyone interested in further details can contact me (Krishnan) as corresponding author.

REVIEWER COMMENTS

Reviewer #3 (Remarks to the Author):

This paper is an introduction to the impact of spacial localisation on the dynamics of biological networks. It includes a number of examples of how simpler ODE approaches could fail to capture the complexity introduced by spatial localisation.

The paper highlights a new level of complexity (to the already complex view) of cellular dynamics. In doing so, it also introduced a number of new open questions (probably beyond the scope of this paper). One open question is how to deal with this complexity in real life? First, there is the amount of data required to model such systems. As pointed by reviewer 1, many examples in the paper are 1d. How would we model even relatively simple pathways? One would need to measure spacial steady state (or time-series for dynamical models) concentrations of all involved species. Second, the number of complex behaviours would grow really fast with the introduction of new species. Observing these behaviours are likely to require quite difficult experiments. Especially at the genome level. Third, fitting parameters to detailed and complex models would lead to high computational complexity, mostly related to the nature of PDEs. Fourth, there are interesting identifiability questions from this work. For example, how to distinguish between similar outcomes from different network/spacial architectures? How much data or experiments are necessary to guarantee identifiability? Perhaps the authors could discuss these (and others) topics in discussion? Section 3 is simply a summary of the paper, and there is little discussion on the implications of this work. I would suggest instead to include a discussion section.

It would have been interesting to have more real life examples, especially in pater formations. Many organisms exhibit pater formations, and systems and synthetic biology has been trying to explain and recreate them over the last decade or two. It would have been interesting to connect these works.

One aspect that the authors mention is that "localization could lead to a negative feedback circuit to be inferred as a positive feedback circuit." One question that would be really interesting (at least for me), is whether localisation could lead to wrong causal network inference. For example, in a network where A does not regulate B, is it possible that using only ODE based network inference could lead to a wrong inference, i.e. that A regulates B? But, this wrongly perceived regulation is only there because we did not take into account spacial localisation. Is it possible to build such an example? This would have important implications in network inference.

Points raised by reviewer 2:

The paper is too long and extensive. Somehow, it feels that it follows the natural research timeline. However, for presentation, it would be extremely beneficial if there was a way to transmit the key messages of the paper in a figure or two (like a graphical abstract, for example), which would be easily grasped by a general audience. Currently, key messages are dispersed over 10 extremely dense figures, targeting only the most interested reader. This was a point raised by reviewer 2 that was not really addressed.

I agree that "the reader should not expect one key cause-effect underlying all the results because there is a fundamentally diverse way in which localization impacts networks", but there should still be a way to summarise key mechanisms that lead to different complex behaviours via spacial localisation in one simple figure. Figure 10 does not help so much as it is also very dense, together with its legend.

Point 1: ok.

Point 2: The authors do not provide a framework to analyse biological data. They just show the different types of behaviours that we can obtain with localisation. The authors answer this clearly in the reply. It is not clear that a data analysis framework could be done within the scope of this paper, although that would be the obvious next step: to make it practical and apply it to data. For example, if spacial and time series data were available, how could models be built to predict

complex behaviours (systems biology)? Or provide design tools to build or modify systems to achieve desired properties (synthetic biology)?

Point 3: The examples added by the authors are in the right direction, but the assumptions on parameters and dynamics is still somewhat arbitrary. Hence, it is still a somewhat abstract example using a known Boolean pathway. I believe reviewer 2 had a more concrete biological example in mind, where some or many of the observed phenomena in the previous Figs 1 to 8 were seen in real biological systems. For example, a case study of a pattern formation, from the many existing in the literature could have perhaps been used here. Nevertheless, these two examples are good additions to the paper.

Point 4: The answers to robustness and noise were somewhat vague. Still, I believe the authors did not mean to show that certain behaviours are robust but instead they could be obtained with spatial localisation. I guess robustness and noise cancellation, if needed, could potentially be achieved with additional layers of complexity and feedback.

Point 5: from what I understood, parameters are chosen to illustrate specific spacial behaviours, especially those that cannot be observed (or simply explained) from ODEs. The parameters are not necessarily chosen to represent real life parameters.

Minor points: I would suggest placing the code in an accessible place, such as GitHub, for example.

Reviewer #3 (Remarks to the Author):

This paper is an introduction to the impact of spacial localisation on the dynamics of biological networks. It includes a number of examples of how simpler ODE approaches could fail to capture the complexity introduced by spatial localisation.

We thank the reviewer for a careful reading of the manuscript and for the comments. These are addressed below.

The paper highlights a new level of complexity (to the already complex view) of cellular dynamics. In doing so, it also introduced a number of new open questions (probably beyond the scope of this paper). One open question is how to deal with this complexity in real life? First, there is the amount of data required to model such systems. As pointed by reviewer 1, many examples in the paper are 1d. How would we model even relatively simple pathways? One would need to measure spacial steady state (or time-series for dynamical models) concentrations of all involved species. Second, the number of complex behaviours would grow really fast with the introduction of new species. Observing these behaviours are likely to require quite difficult experiments. Especially at the genome level. Third, fitting parameters to detailed and complex models would lead to high computational complexity, mostly related to the nature of PDEs. Fourth, there are interesting identifiability questions from this work. For example, how to distinguish between similar outcomes from different network/spacial architectures? How much data or experiments are necessary to guarantee identifiability? Perhaps the authors could discuss these (and others) topics in discussion? Section 3 is simply a summary of the paper, and there is little discussion on the implications of this work. I would suggest instead to include a discussion section.

We agree that these are all relevant questions. We make a few points regarding this

- (i) The fact that the examples are in 1d is not an essential restriction, for the most part. 1-D models are already used to study dynamics of species on the membrane, or across a cross section of a cell. As noted earlier (and developed further in Comment 7, in the Methods Section) the analysis in 1-d allows us to study the effect of localization and the interplay with networks and isolate coherent cause-and-effect relationships, this being facilitated by both computational and mathematical analysis. Most of the underlying insights are equally relevant in higher dimensions, though as we note, there could be additional aspects (eg. pertaining to geometry) which emerge in higher dimensions. We note that computational analysis of localization and compartments can be done using computational software such as VCell, which is a useful tool. However starting in such setting is computationally time consuming in terms of detailed analysis, with many underlying patterns and insights likely to be obscured.
- (ii) With regard to dealing with the complexity of cellular networks while bringing in localization: we now have a subsection of the Discussion Section “Localization amidst the complexity of networks” to provide a perspective and concisely discuss some key points. We make a few points in this regard (a) There is a range of ways in which systems (and synthetic) biology engage with networks: small scale models, moderate scale models and large scale models; synthetic biology uses both bottom-up approaches as well as rewiring of actual networks. Furthermore existing spatial studies involve typically small or moderate scale networks (b) For

small scale networks, our approach can be used almost directly; for moderate scale networks, the point is to focus on the key drivers of the networks and use that as a basis for analysis; similar points can be made for bottom-up studies in synthetic biology; for large scale networks, it depends on the extent of spatial data available: if it is then that can be a basis for using network-centric and space - centric approaches to analyze such networks in conjunction with experiments; if spatial data, for the most part is not available, then the consideration of spatial organization may already reveal an alteration in network connections, and further the focus can be on smaller subnetworks which may play key roles. We point out that with larger networks, a fruitful approach would be to break it down into tractable subnetworks where the analysis can be performed. (c) When we consider larger networks the potential of richer and more complex behaviour is indeed higher at the outset. This is true even for the temporal behaviour of (lumped) networks. A key question is which aspects of this extra complexity are actually deployed. In natural biology, networks are evolved to effectively (and robustly) provide certain functions. The evidence from natural biology (with regard to complex information processing modules for instance) is that key aspects of the networks are organized around this. Other aspects of the network while potentially capable of significantly adding to the (qualitative) complexity, in general do not appear to do so (since the goal is the function). We expect similar insights to hold good here. Spatial localization is a very basic aspect of networks, which occurs across cells, and we expect it to play very important roles in crucial places. Thus we believe that building on concrete experimental cellular evidence on one hand, and working with tractable subnetworks on the other, is a fruitful pathway to examining these issues (d) We point out that there are multiple experimental studies focussing on localization, even at a larger scale (eg. references cited, Sarov et al eLife, 2016)

- (iii) With regard to fitting data in large networks yes, the computational complexity significantly does increase substantially. This is why we believe, this is best approached piece-by piece in a “modular” way in the first instance, keeping in mind the nature of the question addressed. In other instances working with moderate scale networks may be sufficient anyway. It is worth noting that existing concrete spatial studies of networks have tended to involve small to moderate networks.
- (iv) Yes, this does bring in parametric identifiability issues as well. Discussing this will involve further detailed analysis which is beyond the scope of the paper. In this regard we add two points: (a) We indicate the need to distinguish between models with different spatial architectures and how by using localization (possibly manipulated dynamically) it may be possible to discriminate between such architectures (b) In response to another question by the reviewer we present further cases of how it is easy to incorrectly infer certain interactions

These various points are summarized in a subsection of the discussion “Localization amidst the complexity of networks” (Section 3.2).

We also mention (at the end of Section 3.1) that localization of species in multiple locations can result in certain network interactions not occurring, thereby altering the effective network

It would have been interesting to have more real life examples, especially in pattern formations. Many organisms exhibit pattern formations, and systems and synthetic biology has been trying to explain and recreate them over the last decade or two. It would have been interesting to connect these works.

We agree that pattern formation is an interesting concrete context to explore. However (and noting the editorial comment that this was optional) we have not presented more studies on pattern formation since (a) The paper already has a broad range of results, including concrete biological problems (b) There are also results directly pertaining to pattern formation (an entire subsection of results). (c) Trying to develop another new context would lengthen the paper and distract the focus, while not having the space to develop such a study properly.

One aspect that the authors mention is that “localization could lead to a negative feedback circuit to be inferred as a positive feedback circuit.” One question that would be really interesting (at least for me), is whether localisation could lead to wrong causal network inference. For example, in a network where A does not regulate B, is it possible that using only ODE based network inference could lead to a wrong inference, i.e. that A regulates B? But, this wrongly perceived regulation is only there because we did not take into account spacial localisation. Is it possible to build such an example? This would have important implications in network inference.

Yes indeed, there are multiple instances of such a possibility. We discuss this concisely in the Discussion, with further details in Supplementary Material (Section 4.3), both reproduced below:

Discussion:

We found that with networks containing nodes with different diffusivities, qualitatively incorrect inferences emerge in very simple cases. There are also multiple instances (by ignoring localization) where an inference of an interaction between two species may be made, where none exists (see Supplementary Material Sec. 4.3 for details). As examples, suppose two components are in different spatial compartments and non-interacting. Working in an ODE framework could result in (i) incorrect inference about their interaction based on in-vitro data or data in another context (ii) A correlated regulation of these two components (eg global regulation) could be misinterpreted as an interaction (iii) The possibility of a common factor functioning in different ways in two compartments (enabled by localization, absent otherwise) and regulating these may be ignored, leading to a misinterpretation of interaction between the two components (an example related to bifunctional enzymes is discussed in Krishnan, 2020, but this applies more broadly). Furthermore, working in an ODE framework can lead to incorrect inferences about the nature of some interactions, which can then directly lead to other postulated interactions (non-existent) inferred, to explain network behaviour. Finally, one node affecting another node’s localization may be inferred incorrectly as an interaction

Supplementary Material:

The main text (Discussion section) presents examples where an interaction may be inferred in a network (by ignoring localization), while none exists. There are multiple examples of this

- (a) A simple example is inferring an interaction based on in-vitro data, but the interaction not occurring in vivo, because the two components are in different locations
- (b) Similarly an interaction may be relevant in some contexts but not others: for instance two components may interact in a certain phase of the cell cycle (when they are in the same spatial compartment) but not in another phase (when they are in different spatial compartments). Here one type of data may be used to infer something about the system (in a slightly different context) which may be incorrect
- (c) Consider two nodes regulated by a common element S. Depending on the data, a direct (or indirect) regulation of A and B by S, may be inferred as an interaction between A and B. If A and B are in different compartments, they may not have any interactions, but if this possibility is ignored, then an incorrect inference may be made.
- (d) Similar conclusions hold good if the interaction between A and B (in different compartments) is indirect (eg through shared global components). Other indirect effects (eg mechanical effects) could be incorrectly inferred as an interaction between A and B, especially if those effects or factors are outside the postulated inference framework.
- (e) Having different compartments allows common components to function in different ways. An example (discussed in Krishnan et al, 2020) is of a bifunctional enzyme which can act (primarily) as a kinase in one compartment and phosphatase in another compartment at the same time. This dual behaviour is directly enabled by different compartments. If the possibility of different compartments is ignored, then such a possibility may not be accounted for. This then could result in an incorrect inference. For instance suppose the signal activates A in one compartment and inhibits B in another compartment, for instance by being a kinase in the first compartment and a phosphatase in the second compartment. Furthermore suppose experiments (based on measurements in the first compartment) establish the kinase activity. Then the observed steady state regulation could be mis-inferred as the signal activates A and A inhibits B.
- (f) A similar conclusion could be made at the cell population level where common factors could have different regulatory properties in different cell types (eg. resistant cell vs wild-type cell) localized in different regions. The assumption of homogeneity would rule that out, and could result in an incorrect inference.
- (g) We have already shown the feature of an “inverse response” wherein one node (A) regulating another (two-diffusivity) node (B) can result in B exhibiting an opposite type of response in certain regions/locations (eg A inhibiting B can result in an elevation of B in certain regions). Suppose B regulates another node C present at such a location. Then by ignoring localization, and working in an ODE framework would result in either (a) inferring an opposite type of regulation of C by B or (ii) postulating another regulation of C to explain the data, which is non-existent.
- (h) We have already shown how by ignoring spatial localization (of interactions), a negative feedback circuit could be inferred as a positive feedback circuit. This then means that if the intermediate component in the feedback loop is regulating another network component, that the nature of the regulation would be incorrectly inferred. Therefore to explain the correct observed behaviour of that other component would likely involve another interaction which would be an incorrect inference.
- (i) We have already seen in Section 2.3 that for certain types of measurements localization can create a qualitative discrepancy between the actual behaviour and that of the ODE (with the same interactions). Having to account for the data in an ODE

framework may then involve the inference of other (non-existing) interactions to make up for this discrepancy,

- (j) We also point out that a component may regulate the localization of another component and this could be incorrectly inferred as a direct interaction. For instance, we have shown how negative feedback circuit (involving a two-diffusivity node) with one interaction localized could be inferred as a positive-feedback circuit. Now suppose an external node X was causing the localization of this interaction. From the data it would appear (for eg by knocking out X) that X was responsible for the bistability, and consequently playing a direct regulatory role.

Points raised by reviewer 2:

The paper is too long and extensive. Somehow, it feels that it follows the natural research timeline. However, for presentation, it would be extremely beneficial if there was a way to transmit the key messages of the paper in a figure or two (like a graphical abstract, for example), which would be easily grasped by a general audience. Currently, key messages are dispersed over 10 extremely dense figures, targeting only the most interested reader. This was a point raised by reviewer 2 that was not really addressed.

I agree that “the reader should not expect one key cause-effect underlying all the results because there is a fundamentally diverse way in which localization impacts networks”, but there should still be a way to summarise key mechanisms that lead to different complex behaviours via spacial localisation in one simple figure. Figure 10 does not help so much as it is also very dense, together with its legend.

We agree with this point. To address this we have replaced the original Figure 10 (which was essentially a tabular summary) with a pictorial representation involving 8 panels, two from each subsection of the results (2.1-2.4). In this manner we cover many of the essential points. We have provided a self-contained caption to allow for a reader to access the main points of the paper

Incidentally we have also removed Panel (E) of Fig. 2 as we felt that was distracting the flow, without making an important enough point.

Point 1: ok.

Point 2: The authors do not provide a framework to analyse biological data. They just show the different types of behaviours that we can obtain with localisation. The authors answer this clearly in the reply. It is not clear that a data analysis framework could be done within the scope of this paper, although that would be the obvious next step: to make it practical and apply it to data. For example, if spacial and time series data were available, how could models be built to predict complex behaviours (systems biology)? Or provide design tools to build or modify systems to achieve desired properties (synthetic biology)?

Yes, a data analysis framework is outside the scope of the current paper with many issues of its own. That would be a logical next step, for which the results of the paper serve as a foundation, and that needs a dedicated detailed study of its own.

Point 3: The examples added by the authors are in the right direction, but the assumptions on parameters and dynamics is still somewhat arbitrary. Hence, it is still a somewhat abstract example using a known Boolean pathway. I believe reviewer 2 had a more concrete biological example in mind, where some or many of the observed phenomena in the previous Figs 1 to 8 were seen in real biological systems. For example, a case study of a pattern formation, from the many existing in the literature could have perhaps been used here. Nevertheless, these two examples are good additions to the paper.

We are pleased to note that the Reviewer regards the existing two examples as good additions to the paper. We make a comment regarding the parameters. In the case of gradient transduction through a cascade, we identify a key qualitative difference between two potential “wirings” of the cascade (both consistent with data), and show a key qualitative difference between the two: this can be traced to the nature of the regulation in each instance (which we have revealed analytically) and this aspect is independent of exact parameter values. With regard to the cell cycle examples, we focus our models, starting with postulations in the literature (based on a number of experiments) and address key questions in the biological literature, which are of a qualitative nature. For this purpose we need to address whether a certain type of behaviour does indeed emerge from the circuit, what the consequences are, and what the effect of other regulatory interactions are. We are able to address the questions at this level without requiring a detailed analysis of parameters (focussing on the network structure and qualitative trends) We have now included a paragraph while discussing the parameters in these sections of the Models stating that the parameter choices for our analysis is sufficient for our study.

Point 4: The answers to robustness and noise were somewhat vague. Still, I believe the authors did not mean to show that certain behaviours are robust but instead they could be obtained with spatial localisation. I guess robustness and noise cancellation, if needed, could potentially be achieved with additional layers of complexity and feedback.

With regard to robustness, what we have intended to show was that the behaviour we demonstrated could readily be obtained across various parameter sets, depending as it did, on the interplay of key features in the network. We meant that the results could be robustly obtained.

Indeed if in an actual cellular context if robust functioning of networks and buffering against noise was required, that could be done with additional layers of complexity and feedback

Point 5: from what I understood, parameters are chosen to illustrate specific spacial behaviours, especially those that cannot be observed (or simply explained) from ODEs. The parameters are not necessarily chosen to represent real life parameters.

What we had meant to say was this: we start with a basal behaviour in a network (eg bistability, which may occur in a certain range), then choose a representative parameter set in the (“middle” of the) range, i.e. not close to edge of the range. With that, the emergent behaviour follows, in almost all cases for clear and transparent reasons. So we did not rely on a particular parameter set for the behaviour to manifest. The behaviour emerges as a consequence of the network, the basal behaviour and the way localization emerges (for a reason which is transparent and in many cases established analytically or is obtained constructively), and will be present in a range of associated parameters. Thus there is flexibility in choosing these parameters.

In some cases where we show unexpected or unusual behaviour precluded in the ODE, where the existence of the behaviour itself is notable (eg. tristability in a network capable only of exhibiting bistability, but not tristability in the ODE), a single parameter set suffices to make the point (regarding the relaxation of constraints).

The parameters are not necessarily chosen to represent real-life parameters, but the behaviour could indeed be observed in associated networks whose parameters may be in ranges observed in cells in specific contexts.

Minor points: I would suggest placing the code in an accessible place, such as GitHub, for example..

This has been done

REVIEWERS' COMMENTS

Reviewer #3 (Remarks to the Author):

The authors have addressed all the points raised in my review and in the original review of reviewer 2. The discussion section has significantly improved, and it now contains multiple directions of future work. Finally, Figure 10 summarises the paper in one figure, which can be used initially by a reader to get an overview of the paper. It is still a dense figure, but I cannot see another easier way to convey the key messages. I would just suggest that this figure could come earlier (basically, not being the last figure, but instead the first).

Reviewer #3 (Remarks to the Author):

The authors have addressed all the points raised in my review and in the original review of reviewer 2. The discussion section has significantly improved, and it now contains multiple directions of future work. Finally, Figure 10 summarises the paper in one figure, which can be used initially by a reader to get an overview of the paper. It is still a dense figure, but I cannot see another easier way to convey the key messages. I would just suggest that this figure could come earlier (basically, not being the last figure, but instead the first).

We thank the reviewer for a reading of the revised manuscript and for the comments. We thank the reviewer for the constructive suggestion. We did consider this possibility, and understand the reason for the reviewer suggesting it. However, we believe that this figure is best placed at the end of the paper, rather than the beginning since (i) The figure has many details which would not be comprehensible to a reader without an acquaintance of the basic set up (ii) Fig. 1 (c) already provides a schematic summary of the type of insights which the paper will draw out.

Keeping the reviewer's point in mind, we have added a phrase in the captions for Fig. 1(c), indicating that the results are summarized in Fig. 10. By doing so, we indicate the presence of a summary of results and where it may be located, for a reader who may want to jump straight to the summary. However we do not distract other readers' focus as they are reading the paper.